# CFT and Lattice Correlators Near an RG Domain Wall between Minimal Models

Cameron V. Cogburn[a1], A. Liam Fitzpatrick[a2], Hao Geng[b3]

[a]*Department of Physics, Boston University, Boston, MA 02215, USA*
[b]*Jefferson Physical Laboratory, Harvard University, Cambridge, MA 02138, USA*

## Abstract

Conformal interfaces separating two conformal field theories (CFTs) provide maps between different CFTs, and naturally exist in nature as domain walls between different phases. One particularly interesting construction of a conformal interface is the renormalization group (RG) domain wall between CFTs. For a given Virasoro minimal model $\mathcal{M}_{k+3,k+2}$, an RG domain wall can be generated by a specific deformation which triggers an RG flow towards its adjacent Virasoro minimal model $\mathcal{M}_{k+2,k+1}$ with the deformation turned on over part of the space. An algebraic construction of this domain wall was proposed by Gaiotto in [1]. In this paper, we will provide a study of this RG domain wall for the minimal case $k = 2$, which can be thought of as a nonperturbative check of the construction. In this case the wall is separating the Tricritical Ising Model (TIM) CFT and the Ising Model (IM) CFT. We will check the analytical results of correlation functions from the RG brane construction with the numerical density matrix renormalization group (DMRG) calculation using a lattice model proposed in [2, 3], and find a perfect agreement. We comment on possible experimental realizations of this RG domain wall.

---

[1]Email: cogburn@bu.edu
[2]Email: fitzpatr@bu.edu
[3]Email: haogeng@fas.harvard.edu

# 1 Introduction and Summary

A useful concept in the study of Quantum Field Theory (QFT) is the idea of a 'space of QFTs' [4]. If we view QFTs as Renormalization Group (RG) flows between Conformal Field Theory (CFT) fixed points, then the space of QFTs can be envisioned as a network of CFT points connected to each other by paths along which a family of quantum field theories are defined interpolating between an ultraviolet (UV) CFT and an infrared (IR) CFT. These QFTs are parameterized by the energy scale along the RG flow. In general, the dynamics along such flows is complicated and does not benefit from the relative rigidity of its CFT endpoints. Instead of studying the full RG flows themselves, an appealing construction is that of 'RG branes' (aka RG Domain Walls), which capture much of the information of an RG flow but in a simpler setting. RG branes are constructed by taking a relevant deformation that triggers the RG flow from the UV CFT to the IR CFT and turning it on over only part of space, so that in the infrared regime one obtains the 'IR CFT' in one spatial region and the 'UV CFT' everywhere else. The boundary between these two regions is the RG brane, which thereby collapses the entire RG flow to the geometric action of moving across this boundary. Moreover, symmetric choices of the boundary can preserve a subset of the conformal symmetries of the CFT endpoints.

Another advantage of RG branes is that they are relatively easy to engineer in practice. In the context of numerical simulations, for instance with a computation on a lattice, one simply has to choose parameters in the underlying theory to pick out the UV CFT on half of the space and the IR CFT on the rest of the space. In fact, this approach will be one of our main tools for studying RG branes in this paper. Moreover, this technique translates into a well-defined protocol for creating an RG brane experimentally, assuming one has the flexibility to tune to a critical point over only part of space.

Our main goal in this paper will be to compute physical observables in an interesting class of RG branes. We will focus on two-point correlation functions, partly for simplicity but also partly with an eye towards the potential connection to experimental measurements in the future. In CFTs, two-point functions in the presence of a boundary are comparable in complexity to four-point functions in its absence, so the theoretical calculation of them provides the opportunity to predict a fairly complicated set of observables that might be measurable in practice.

We will focus on a specific RG brane scenario that brings together two remarkable pieces of work, one from a CFT perspective and one from an underlying microscopic perspective. The first of these is a recent proposal [2, 3] for an experimental realization, as well as an explicit lattice Hamiltonian describing it, of a supersymmetric quantum critical point that can be obtained by tuning only a single parameter. The specific instance of this proposal that we will use produces a phase diagram with two distinct phases, one gapped and one gapless, separated by a critical point. Moreover, the gapless phase is described in the IR by the 2d Ising Model CFT, and the critical point between the phases is decribed by the Tricritical Ising Model CFT. This lattice Hamiltonian can therefore be used to construct an RG brane

separating the 2d Ising Model (IM) CFT from the Tricritical Ising Model (TIM) CFT, and will give us the first numeric handle for computing observables numerically.

The second result we will use [1] is a proposal for the RG brane between consecutive Virasoro minimal models, directly within the CFT description.[4] The first two Virasoro minimal models are exactly the Ising Model CFT and the Tricritical Ising Model CFT, with central charges $c = \frac{1}{2}$ and $c = \frac{7}{10}$ respectively. This proposal will give us a second handle for computing observables, this time directly in the IR limit. In particular, we will compute two-point functions of operators in the presence of the RG brane using the CFT proposal of [1], and compare to computations using a DRMG analysis of the lattice Hamiltonian from [3]. We will find remarkably good agreement.

The paper is organized as follows. In section 2 we spell out how the RG brane in our paper in constructed on the lattice. In section 3, we compute several energy-energy correlators near the RG domain wall between the Tricritical Ising Model CFT and the Ising Model CFT. In section 4, we compare our analytical results in 3 with the numerical results obtained from the lattice construction in 2. In section 5, we discuss potential experimental consequences and other future directions. We review many technical details of the RG brane proposal in [1] in a series of appendices: in section A we review the coset (Sugawara) construction of minimal Virasoro models, and in section B, we review the construction of the RG brane itself. In section C we review some useful properties of topological superconductors relevant to our experimental proposal.

## 2 RG Brane Lattice Construction

### 2.1 General Strategy

We are interested in RG branes connecting a "UV" CFT on one side to an "IR" CFT on the other side. To construct such an RG brane in a lattice model, we need to be able to dial the parameters of the lattice Hamiltonian such that for some parameters, the low energy limit of the lattice model is described by the UV CFT, and for other parameters the low energy limit is described by the IR CFT. The "UV" and "IR" labels of the two CFTs in this context are solely relative to each other, as both of them describe the physics at infrared scales compared to the underlying lattice spacing. In order for the UV CFT to be able to have an RG flow to the IR CFT, there must exist a relevant deformation of the UV CFT that triggers this RG flow. In the space of lattice parameters, this means that points describing the UV CFT require tuning (at least) one parameter to a critical point (or surface). Moreover, it is then possible to detune away from this point in some direction such that the low energy limit is described by the IR CFT. We can choose to parameterize this direction as a coupling $h$ in our Hamiltonian:

$$H = H_1 + hH_2, \tag{2.1}$$

---

[4]See [5–7] for the extension of the construction to other 2d CFTs and [8] for an interesting construction of a domain wall between different symmetry-protected topological (SPT) phases in the same spirit.

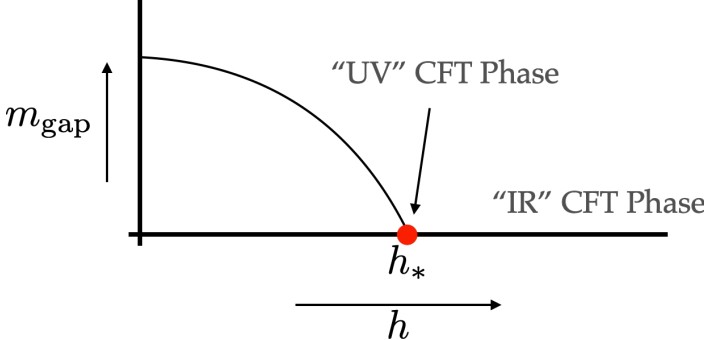

**Figure 1**. Phase diagram required for the construction of the RG brane. At a critical value $h_*$ of the coupling, the low-energy limit is described by the "UV" CFT, whereas for $h > h_*$ it is described by the "IR" CFT. The RG brane is constructed by tuning $h$ to $h_*$ on half of the space, and taking $h$ slightly greater than $h_*$ on the other half of the space. The specific behavior at $h < h_*$ is irrelevant for our discussion.

such that $h = h_*$ is the critical value that produces the UV CFT at low energies, and $h > h_*$ produces the IR CFT at low energies. This setup is depicted in Fig. 1.

Given a Hamiltonian with these properties, it is then straightforward to construct the RG brane in terms of the lattice theory: one simply has to tune $h = h_*$ on part of the space, and take $h > h_*$ on the rest of the space,[5] and the RG brane lies on the boundary separating these two regions.

## 2.2 Lattice Model for Tricritical Ising to Ising RG Brane

Although the general setup described in the previous subsection should be possible to be realized in many different examples, in this work we will focus mostly on the specific lattice system proposed in [2, 3]. Following their notation, the Hamiltonian is $H = H_1 + hH_2$ with

$$H_1 = \sum_i \left[ \sigma_i^z \sigma_{i+1}^z + \sigma_i^x + \mathcal{J} \left( \mu_{i,a}^z \mu_{i,b}^z + \mu_{i,b}^z \mu_{i+1,a}^z \right) + g \left( \sigma_i^x \mu_{i,a}^z - \sigma_i^z \sigma_{i+1}^z \mu_{i,b}^z \right) \right],$$

$$hH_2 = -h \sum_i (\mu_{i,a}^x + \mu_{i,b}^x),$$

(2.2)

where $\mu_{i,a}, \mu_{i,b}$ and $\sigma_i$ are all Pauli matrices for spin-1/2 spins. The virtue of this model is that, due to an underlying symmetry of the theory, only one tuning is necessary in order to reach the (supersymmetric) Tricritical Ising Model fixed point.

---

[5]One should not choose $h$ to be too close to or too far from $h_*$. If $h - h_*$ is too small, then this corresponds to setting a very tiny coefficient in front of the relevant deformation of the UV CFT, so that the IR CFT is only reached at very long distances. For an infinite sized lattice, this would not present any problem, but in a finite-sized system it would prevent one from reaching the IR CFT. On the other hand, if $h - h_*$ is taken to be too large, then it is no longer clear that detuning $h$ away from $h_*$ can be described as a local relevant deformation of the UV CFT.

To roughly understand why, as explained in [3], first note that if we neglect the terms proportional to $\mathcal{J}, g$, and $h$, the Hamiltonian is just that of the critical Ising Model,

$$H_* = \sum_i \sigma_i^z \sigma_{i+1}^z + \sigma_i^x, \tag{2.3}$$

which describes a gapless free fermion. The coupling between $\sigma$ and $\mu$ in general could gap out the system, but does not as long as $h$ is sufficiently large. The reason is that at large (positive) $h$, the $\mu$ spins do not order along the $z$ direction, in which case the vanishing mass gap is protected by an unbroken $\mathbb{Z}_2$ symmetry that flips the sign of $\mu^z$ ($\mu_{i,a}^z \to -\mu_{i,b}^z$ and $\mu_{i,b}^z \to -\mu_{i+1,a}^z$) and acts nontrivially (essentially, as the high-temperature/low-temperature duality map) on $\sigma_i$ ($\sigma_i^z \sigma_{i+1}^z \leftrightarrow \sigma_i^x$). As $h$ is decreased, eventually there is a phase transition to the ordered phase where $\langle \mu^z \rangle \neq 0$. At this critical point, the order parameter behaves as a massless scalar degree of freedom, allowing for a supersymmetric spectrum. For smaller values of $h$, the theory is gapped.

The upshot is that for any value of model parameters $\mathcal{J}$ and $g$, there is a critical value of $h$ where the low-energy limit is described by the Tricritical Ising Model, and for larger values of $h$ the low energy limit is described by the critical Ising Model CFT. The authors of [2] found that for the specific choice of $\mathcal{J} = 1, g = \frac{1}{2}$, this critical value is approximately $h_* = 1.62$. [6]

In field theory language, the theory can be described as a single chiral superfield $\Phi$ with a superpotential given by

$$W(\Phi) = \kappa\Phi + r\Phi^3. \tag{2.4}$$

Supersymmetry is broken spontaneously if $W'(\Phi) \neq 0$. Semiclassically, at $\frac{r}{\kappa} < 0$ the vacuum has $\langle \Phi \rangle \neq 0$ and $W'(\Phi) = 0$, so supersymmetry is preserved but a $\mathbb{Z}_2$ symmetry under which $\Phi$ changes sign is spontaneously broken and the theory is gapped; by contrast, at $\frac{r}{\kappa} > 0$, $\langle \Phi \rangle = 0$ and $W'(\Phi) \neq 0$, so the supersymmetry of the theory is broken spontaneously, producing a massless fermionic Goldstino, which is the massless Majorana fermion of the $c = 1/2$ critical Ising Model. This semiclassical picture continues to hold at the quantum level (see e.g. [11]), and the critical point between the two phases is again described by the Tricritical Ising Model CFT.

We numerically solve for the RG brane using a Density Matrix Renormalization Group (DMRG) method based on the iTensor library [12, 13]. This DMRG algorithm is controlled by a few key parameters, as explained in the documentation. For our simulations we found the parameter values of `nsweeps` = 15, `maxdim` = $[10, 20, 100, 175, 250]$, and `cutoff` = 1E-12 were sufficient for even large (eg. $N = 300$) lattices. In setting up the lattice, periodic boundary conditions are enforced. Therefore the spacetime geometry is such that we have a tube, the boundary of which is a circle. On this circle the RG brane is placed at 0 and 180 degrees, with the Ising model ($h = 2$) on one side and the Tricritical Ising Model ($h = 1.62$) on the other.

---

[6]We notice that the O'Brien-Fendley lattice model [9, 10] provides another example where we can drive a system from the Ising fixed point to the Tricritical Ising fixed point by tuning a single parameter.

Discretizing a system on a lattice inherently introduces lattice spacing and finite volume effects. In Sec. 4.1 we expand the $\mu$ lattice operator in terms of the CFT operators, the coefficients of which are found by computing the finite volume two-point function when the lattice is entirely in either the Ising CFT phase or Tricritical Ising CFT phase. These coefficients precisely determine the overall normalization between the numerical lattice calculation and the analytic CFT calculation, the result of which is shown in Sec. 4.2.

## 3  The Computation of Correlators In the Presence of the RG Brane

In this section, we extend the study of the RG domain wall in [1] to the computation of specific four-point functions (in the unfolded picture). These four-point functions reduce to two-point functions of composite operators in the folded description. We firstly spell out our general strategy in computing the relevant four-point functions using Gaiotto's construction of the RG domain wall. We then explicitly consider the case of the RG domain wall of the Tricitical Ising Model. We compute two such four-point functions following our strategy, compare the results with numerical calculation by DMRG and find precise agreement.

A priori, it is not at all obvious that one should be able to do analytic computations of correlation functions in the presence of the RG brane. In general, even if the UV and IR CFTs are rational, so that all their correlators can be computed analytically, it will not necessarily be the case that the RG brane between them is also rational, and one might have to accept that the resulting correlators cannot be computed based on the algebra alone. A class of interfaces that do preserve the individual algebras of the UV and IR CFTs are called *rational interfaces*. From this point of view, what is special about rational interfaces is that they satisfy boundary conditions that glue the chiral sector of the algebra of the theory $\overline{\mathcal{T}}_L \times \mathcal{T}_R$ to the anti-chiral sector by an automorphism (i.e. the map preserves the commutation relations between the symmetry generators). With the automorphism specified the rational interfaces can be classified by Cardy's algebraic description. However, there is an inherent tension between preserving the UV and IR CFT algebras and simultaneously coupling them together in a nontrivial way. The reason is that the UV and IR CFT algebras strongly constrain the allowed form of the correlators of the theory, and in fact the reason why rational CFTs are analytically tractable is that the algebra almost completely determines their correlators. Consequently, if the algebra completely factorizes into the UV and IR CFT algebras, then their correlators also factorize into UV and IR CFT correlators. In more technical terms, the automorphism that maps the chiral and anti-chiral algebras into each other across the RG brane usually doesn't mix the $L$ and $R$ sectors of the algebra. In order to mix them in a nontrivial way, there must be a hidden symmetry that relates the $L$ and $R$ sectors. Naively, our setup has no hidden symmetry and therefore one might expect that the RG brane is not related to any rational interface. Fortunately, this expectation is too pessimistic, and Gaiotto [1] has proposed a remarkably elegant construction of RG interfaces between consecutive 2d Virasoro minimal model CFTs using rational interfaces by identifying the RG branes as the rational interfaces of a different 2d CFT than $\overline{\mathcal{T}}_L \times \mathcal{T}_R$ which contains a larger

symmetry algebra. This construction is based on the observation that the larger symmetry algebra is $\widetilde{\mathcal{B}}$ (that we review in detail in Sec. A.2) and it extensively uses results reviewed in App. A. The basic observation that leads to the realization of $\widetilde{\mathcal{B}}$ as the proper algebra is from the considerations of topological defects [14] and the perturbative results of the RG flow [15, 16]. We also review Gaiotto's algebraic proposal for the RG brane in detail in App. B. The proposal can be roughly summarized as the statement that the RG brane is simply the product of a conformal interface that maps operators from the tensor product theory to the 'B' theory with the enhanced $\widetilde{\mathcal{B}}$ chiral algebra, times a Cardy boundary condition with respect to this algebra.

## 3.1 The General Strategy

Our main goal is to compute correlation functions of local operators in the presence of the RG brane between nearest neighbor minimal models, $\mathcal{M}_{k+3,k+2}$ and $\mathcal{M}_{k+2,k+1}$. We will mostly focus on the case of the RG brane between TIM and Ising (i.e. $k = 2$), though the generalization to other minimal models is conceptually straightforward. The simplest kind of two-point function one might try to consider is that of two operators in, say, $\mathcal{M}_{k+2,k+1}$:[7]

$$\langle \phi_{r,s}^{(k+1)}(x_1, \overline{x_1}) \phi_{r,s}^{(k+1)}(x_2, \overline{x_2}) \rangle_{RG}. \tag{3.2}$$

However, it will be useful to instead start with correlators where each local operator is a product of one operator from $\mathcal{M}_{k+2,k+1}$ and one from $\mathcal{M}_{k+3,k+2}$:

$$\langle (\phi_{r,s}^{(k+1)} \phi_{s,t}^{(k+2)})(x_1, \overline{x_1}) (\phi_{r,s}^{(k+1)} \phi_{s,t}^{(k+2)})(x_2, \overline{x_2}) \rangle_{RG}. \tag{3.3}$$

When $(s,t) = (1,1)$, this reduces to the former type of two-point function (and similarly if $(r,s) = (1,1)$). However, our strategy for computing correlators will be first to map operators from the tensor product, or "$\mathcal{A}$", theory $\mathcal{M}_{k+2,k+1} \times \mathcal{M}_{k+3,k+2}$ (which arises from the folded description) into operators in the "$\widetilde{\mathcal{B}}$" theory, which can be described in terms of the Ising model times the $(k-1)$-th supersymmetric minimal model $\mathcal{SM}_{k+1,k+3}$. For $k = 2$, both descriptions are (loosely speaking) TIM $\times$ Ising, but the mapping of operators is still nontrivial.

We will work out in detail the two-point function of $\epsilon$ from Ising and the two-point function of $\epsilon$ from TIM. These operators are represented in the $\widetilde{\mathcal{B}}$ theory as follows [17]:

$$(\epsilon^{TIM})_A \propto (\sigma^{IM} \sigma^{TIM})_B, \qquad (\epsilon^{IM})_A \propto (\sigma^{IM} \sigma'^{TIM})_B. \tag{3.4}$$

More generally, the strategy of computing such two-point function is the following:

---

[7]The individual operators in TIM and Ising sit in the Kac table as follows:

$$1^{IM} = \phi_{(1,1)}^{IM} = \phi_{(2,3)}^{IM}, \quad \epsilon^{IM} = \phi_{(2,1)}^{IM} = \phi_{(1,3)}^{IM}, \quad 1^{TIM} = \phi_{(1,1)}^{TIM} = \phi_{(3,4)}^{TIM},$$

$$\epsilon^{TIM} = \phi_{(1,2)}^{TIM} = \phi_{(3,3)}^{TIM}, \quad \epsilon'^{TIM} = \phi_{(1,3)}^{TIM} = \phi_{(3,2)}^{TIM}, \quad \epsilon''^{TIM} = \phi_{(1,4)}^{TIM} = \phi_{(3,1)}^{TIM}. \tag{3.1}$$

1. Use the branching rule to map the composite operator $(\phi_{r,s}^{(k+1)}\phi_{s,t}^{(k+2)})(x_1,\overline{x_1})$ from the $\mathcal{M}_{k+2,k+1}\times\mathcal{M}_{k+3,k+2}$ description to $[r,d';t]\otimes[d,\widetilde{d};d']$ in the $\mathcal{SM}_{k+3,k+1}\times\mathcal{M}_{4,3}$ description.[8]

2. Do the bulk OPE expansion for $\mathcal{SM}_{k+3,k+1}$ and $\mathcal{M}_{4,3}$ separately to have $[r,d';t]\times[r,d';t]\to[\underline{r},\underline{d'};\underline{t}]$ and $[d,\widetilde{d};d']\otimes[d,\widetilde{d};d']\to[\underline{d},\underline{\widetilde{d}};\underline{d'}]$ with known conformal block and OPE coefficients.

3. Project the $[\underline{r},\underline{d'};\underline{t}]\otimes[\underline{d},\underline{\widetilde{d}};\underline{d'}]$'s which can not be mapped back to the OPE spectrum of $\phi_{r,s}^{(k+1)}\phi_{s,t}^{(k+2)}$ with itself in the $\mathcal{M}_{k+2,k+1}\times\mathcal{M}_{k+3,k+2}$ description.

4. Look for the boundary operator expansion (BOE) coefficient of $[\underline{r},\underline{d'};\underline{t}]\otimes[\underline{d},\underline{\widetilde{d}};\underline{d'}]$ from the exact Cardy state Equ. (B.7).

5. Multiply the BOE coefficients with the corresponding bulk conformal blocks and bulk OPE coefficients, and sum over all possible $\underline{r},\underline{d},\underline{d'},\underline{t}$ and $\underline{\widetilde{d}}$.

This procedure fixes the final correlator up to an overall proportionality constant. This constant comes from the mapping in the first step, and can be fixed by the one-point function calculated using the method in Sec. B.5.[9] Alternatively, the overall coefficient can be determined simply by fixing the leading OPE singularity of the two-point function.

In general, the prediction for the correlators in the presence of the RG brane will take the form

$$
\langle(\phi_{r,s}^{(k+1)}\phi_{s,t}^{(k+2)})(x_1,\overline{x_1})(\phi_{r,s}^{(k+1)}\phi_{s,t}^{(k+2)})(x_2,\overline{x_2})\rangle_{RG}
$$
$$
=\sideset{}{'}\sum_{i,j}\langle(\phi_i^{(k+1)}\phi_j^{(k+2)})_B|RG\rangle C_{(r,s),(r,s),i}^{(k+1)}C_{(s,t),(s,t),j}^{(k+2)}G_{(r,s),(r,s),i}^{(k+1)}(z)G_{(s,t),(s,t),j}^{(k+2)}(z),
\tag{3.5}
$$

where the $C$s and $G$s are OPE coefficients and conformal blocks in the 'B' theory, respectively, the conformal cross ratio $z$ is defined below, and the prime on the sum indicates that certain terms in the sum are discarded.

## 3.2 $\langle\epsilon^{TIM}\epsilon^{TIM}\rangle_{RG}$

First, we will calculate the two-point function of the TIM $\epsilon$ operator in the presence of the RG brane, using the branching rule we have $(1^{IM}\epsilon^{TIM})_A \propto (\sigma^{IM}\sigma^{TIM})_B$ from the $\mathcal{A}$ to the $\widetilde{\mathcal{B}}$ theory. Therefore, using the folded description, our task is to compute the following

---

[8]This mapping, though nontrivial, is constrained by preserving total conformal weight and the fact that $r+t$ even lives in the NS sector of the $\mathcal{SM}_{k+3,k+1}$ and odd in the R sector; in subsection 3.4 we demonstrate how to use this constraint to work out the branching map for $\epsilon'^{TIM}$.

[9]Note that there are operators for which the overall constant cannot be fixed by the last step. These are operators $\phi_{r,s}^{(k+1)}\phi_{s,t}^{(k+2)}$ whose one-point function is zero (for example $r+t$ is odd).

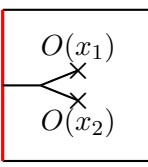

**Figure 2**. *An illustration of the calculation where $O(x) \equiv 1^{IM} \times \epsilon^{TIM}(x)$ and we do the bulk OPE first then compute the resulting one point function due to the appearance of the Cardy boundary.*

correlator in the $\widetilde{\mathcal{B}}$-theory

$$
\begin{aligned}
&\langle (\sigma^{IM}\sigma^{TIM})(x_1, \overline{x}_1)(\sigma^{IM}\sigma^{TIM})(x_2, \overline{x}_2) \rangle_{RG} \\
&= \langle (\sigma^{IM}\sigma^{TIM})(x_1)(\sigma^{IM}\sigma^{TIM})(\overline{x}_1)(\sigma^{IM}\sigma^{TIM})(x_2)(\sigma^{IM}\sigma^{TIM})(\overline{x}_2) \rangle .
\end{aligned} \tag{3.6}
$$

The correlator in the unfolded description is a sum over four-point conformal blocks, where the four external operators are $(\sigma^{IM}\sigma^{TIM})$ as written above. These conformal blocks are simply products of the conformal blocks of $\sigma^{IM}$ and $\sigma^{TIM}$ in the Ising model and TIM, respectively, which are known in closed form (see below). To compute the coefficient of each conformal block, we first perform the operator production expansion (OPE) in the bulk, which is inherited from the individual Ising and TIM factors:

$$
[\sigma^{IM}]\times[\sigma^{IM}] = [1^{IM}]+[\epsilon^{IM}], \quad [\sigma^{TIM}]\times[\sigma^{TIM}] = [1^{TIM}]+[\epsilon^{TIM}]+[\epsilon'^{TIM}]+[\epsilon''^{TIM}]. \tag{3.7}
$$

The OPE of the products of chiral components of the fields in the $B$ theory is, roughly speaking, a sort of "square root" of the products of OPE of the non-chiral fields from the Ising Model and the Tricritical Ising Model, as explained in [17] and described in more detail below. Finally, to obtain the coefficient of the conformal block, we perform the bulk-to-boundary expansion of each operator in the bulk OPE, which simply contributes an additional factor of the bulk operator's one-point function. This OPE,

$$
\langle \mathcal{O}(x) \times \mathcal{O}(y) \rangle_{RG} = \sum_i C_{\mathcal{O}\mathcal{O}i}(x,y)\langle \mathcal{O}_i(y) \rangle_{RG}, \tag{3.8}
$$

is depicted in Fig.2. In principle, we could evaluate the above expression by doing a brute-force numerical sum over all operators $\mathcal{O}_i$ in the bulk OPE of $\epsilon^{TIM} \times \epsilon^{TIM}$ in the tensor product theory. The advantage of using the full chiral algebra of the $\widetilde{\mathcal{B}}$ theory is that there are only a finite number of primary operators, and so by computing the full conformal blocks we reduce the sum to a finite number of terms. A useful consistency check that we will perform of our method is that it reproduces the prediction we get for these terms purely from using the standard bulk OPE for $\epsilon^{TIM}$ in TIM, times the RG brane one-point functions, and in fact we will check enough terms this way to independently fix all the coefficients of the $\widetilde{\mathcal{B}}$ theory conformal blocks.

To evaluate the correlator (3.6), we need to use the OPE of the chiral components $(\sigma^{IM}\sigma^{TIM})$ in the $\widetilde{\mathcal{B}}$ theory, in order to sum over the corresponding conformal blocks. As

explained in [17],[10] this OPE is not just a simple product of the OPEs of the scalar $\sigma^{IM} \times \sigma^{IM}$ and $\sigma^{TIM} \times \sigma^{TIM}$ OPEs in the IM and TIM theories, but instead involves both a projecting-out of certain cross-products between the two OPEs as well as taking a square root of the non-holomorphic OPE coefficients. Of the eight operators one would naively get in the product of the two OPEs in (3.7), four get projected out. A convenient way to infer this is that the OPE must be consistent with the $\mathcal{A}$ theory OPE $\epsilon^{TIM} \times \epsilon^{TIM} \sim 1^{TIM} + \epsilon'^{TIM} + \text{descendants}$, and therefore only products of operators where the total dimension of the product is equal that of $1^{TIM}$ or $\epsilon'^{TIM}$ plus integers can appear. As a result, only conformal blocks corresponding to the appearance of $1^{IM}1^{TIM}, 1^{IM}\epsilon'^{TIM}, \epsilon^{IM}\epsilon^{TIM}$ and $\epsilon^{IM}\epsilon''^{TIM}$ in the $\sigma^{IM}\sigma^{TIM} \times \sigma^{IM}\sigma^{TIM}$ OPE survive. The correlator is given by the following formula

$$\langle(\sigma^{IM}\sigma^{TIM})(x_1, \overline{x}_1)(\sigma^{IM}\sigma^{TIM})(x_2, \overline{x}_2)\rangle_{RG} = \frac{1}{|x_1 - x_2|^{2/5}}G_{\epsilon\epsilon}^{TIM}\left(\left|\frac{x_1 - x_2}{x_1 - \overline{x}_2}\right|^2\right),$$

$$G_{\epsilon\epsilon}^{TIM}(z) \equiv z^{1/5}\sideset{}{'}\sum_{\substack{\phi_i = 1, \epsilon \\ \phi_j = 1, \epsilon, \epsilon', \epsilon''}} \langle(\phi_i^{IM}\phi_j^{TIM})_B|RG\rangle C_{\sigma\sigma\phi_i}^{IM}C_{\sigma\sigma\phi_j}^{TIM}G_{\sigma\sigma\phi_i}^{IM}(z)G_{\sigma\sigma\phi_j}^{TIM}(z). \tag{3.9}$$

where the prime on the sum indicates that only the products of conformal blocks described above are included. The Ising model conformal blocks $G_{\sigma\sigma\phi_i}^{IM}(z)$ are

$$G_{\sigma\sigma1}^{IM}(z) = \frac{1}{\sqrt{2}(z(1-z))^{1/8}}\sqrt{1 + \sqrt{1-z}}, \qquad G_{\sigma\sigma\epsilon}^{IM}(z) = \frac{\sqrt{2}}{(z(1-z))^{1/8}}\sqrt{1 - \sqrt{1-z}}, \tag{3.10}$$

and the TIM conformal blocks $G_{\sigma\sigma\phi_i}^{TIM}(z)$ are [18]

$$G_{\sigma\sigma1}^{TIM}(z) = \frac{1}{z^{3/40}}\left(\frac{\sqrt{\sqrt{1-z}+1}\,{}_2F_1\left(-\frac{2}{5}, \frac{1}{5}; \frac{2}{5}; z\right)}{\sqrt{2}(1-z)^{3/40}} + \frac{z(1-z)^{1/40}\,{}_2F_1\left(\frac{1}{5}, \frac{4}{5}; \frac{7}{5}; z\right)}{2\sqrt{2}\sqrt{\sqrt{1-z}+1}}\right),$$

$$G_{\sigma\sigma\epsilon}^{TIM}(z) = z^{\frac{1}{40}}\left(\frac{\sqrt{\sqrt{1-z}+1}(1-z)^{1/40}\,{}_2F_1\left(-\frac{1}{5}, \frac{2}{5}; \frac{3}{5}; z\right)}{\sqrt{2}} + \frac{\sqrt{2}z(1-z)^{21/40}\,{}_2F_1\left(\frac{4}{5}, \frac{7}{5}; \frac{8}{5}; z\right)}{3\sqrt{\sqrt{1-z}+1}}\right)$$

$$G_{\sigma\sigma\epsilon'}^{TIM}(z) = z^{\frac{21}{40}}\left(2\sqrt{2}\sqrt{\sqrt{1-z}+1}(1-z)^{21/40}\,{}_2F_1\left(\frac{4}{5}, \frac{7}{5}; \frac{8}{5}; z\right) - \frac{3\sqrt{2}(1-z)^{1/40}\,{}_2F_1\left(-\frac{1}{5}, \frac{2}{5}; \frac{3}{5}; z\right)}{\sqrt{\sqrt{1-z}+1}}\right)$$

$$G_{\sigma\sigma\epsilon''}^{TIM}(z) = z^{17/40}\left(\frac{28\sqrt{2}\,{}_2F_1\left(-\frac{2}{5}, \frac{1}{5}; \frac{2}{5}; z\right)}{\sqrt{\sqrt{1-z}+1}(1-z)^{3/40}} - 14\sqrt{2}\sqrt{\sqrt{1-z}+1}(1-z)^{1/40}\,{}_2F_1\left(\frac{1}{5}, \frac{4}{5}; \frac{7}{5}; z\right)\right). \tag{3.11}$$

---

[10]See for instance, their equation (3.6) and surrounding text.

The relevant OPE coefficients are [19–21]

$$C_{\sigma\sigma1}^{IM} = 1,$$

$$C_{\sigma\sigma\epsilon}^{IM} = \frac{1}{2},$$

$$C_{\sigma\sigma1}^{TIM} = 1,$$

$$C_{\sigma\sigma\epsilon}^{TIM} = \frac{5\sqrt{21\Gamma\left(-\frac{7}{4}\right)}\Gamma\left(\frac{3}{4}\right)^2\Gamma\left(\frac{6}{5}\right)\left(\frac{\Gamma\left(\frac{7}{10}\right)\Gamma\left(\frac{5}{4}\right)}{\Gamma\left(\frac{3}{10}\right)}\right)^{3/2}}{2^{3/10}\pi^2\Gamma\left(\frac{4}{5}\right)}, \tag{3.12}$$

$$C_{\sigma\sigma\epsilon'}^{TIM} = \frac{5}{6}\left(\frac{\Gamma\left(\frac{2}{5}\right)}{\Gamma\left(\frac{1}{5}\right)\Gamma\left(\frac{3}{5}\right)}\right)^{3/2}\sqrt{\Gamma\left(\frac{4}{5}\right)}\Gamma\left(\frac{6}{5}\right),$$

$$C_{\sigma\sigma\epsilon''}^{TIM} = \frac{\sqrt{\frac{\Gamma\left(-\frac{7}{4}\right)}{21}}\Gamma\left(\frac{3}{4}\right)\Gamma\left(\frac{5}{4}\right)^{3/2}\Gamma\left(\frac{7}{4}\right)}{2\pi^2},$$

and the relevant one-point functions (or the boundary operator expansion coefficients) can be found in Equ. (B.7) and Equ. (B.8):

$$\langle(1^{IM}1^{TIM})_B|RG\rangle\rangle = \frac{1}{2}\sqrt{S_{0,0}^{(1)}S_{0,0}^{(3)}} = \omega_-, \qquad \langle(1^{IM}\epsilon'^{TIM})_B|RG\rangle\rangle = \frac{1}{2}\sqrt{S_{0,0}^{(1)}S_{0,2}^{(3)}} = \omega_+,$$

$$\langle(\epsilon^{IM}\epsilon^{TIM})_B|RG\rangle\rangle = -\frac{1}{2}\sqrt{S_{0,0}^{(1)}S_{0,2}^{(3)}} = -\omega_+, \qquad \langle(\epsilon^{IM}\epsilon''^{TIM})_B|RG\rangle\rangle = -\frac{1}{2}\sqrt{S_{0,0}^{(1)}S_{0,0}^{(3)}} = -\omega_-,$$

$$\text{where } \omega_\pm = \frac{1}{2}\left(\frac{5\pm\sqrt{5}}{40}\right)^{\frac{1}{4}}, \quad \text{and } S_{r,s}^{(k)} = \sqrt{\frac{2}{k+2}}\sin\left(\frac{\pi(r+1)(s+1)}{k+2}\right). \tag{3.13}$$

---

As mentioned above, a fully independent calculation of the coefficients of the conformal blocks in (3.9) can be obtained by instead matching its short-distance expansion to a calculation of (3.8) using the standard TIM bulk OPE coefficients together with the RG brane one-point functions. There are four independent coefficients to be fixed, one for each conformal block appearing in the sum. One of these is simply a normalization which we can factor out. The small $z$ expansion of $G^{TIM}(z)$ is

$$G^{TIM}(z) \propto z^{-1/5}\left(1 - \frac{\sqrt{2}(11+5\sqrt{5})^{\frac{1}{10}}\pi^{\frac{3}{4}}\Gamma\left(\frac{2}{5}\right)^{\frac{3}{2}}}{3\sqrt{\Gamma\left(\frac{1}{5}\right)}\Gamma\left(\frac{3}{10}\right)^{\frac{3}{2}}\Gamma\left(\frac{4}{5}\right)}z^{3/5}(1+\frac{3z}{10}) + \frac{3}{280}z^2 + \ldots\right) \tag{3.14}$$

The coefficient of $z^{3/5}$ should be exactly $C_{\epsilon\epsilon\epsilon'}$ times $\frac{\langle(\epsilon'^{TIM})_A\rangle_{RG}}{\langle(1)_A\rangle_{RG}}$. The OPE coefficient $C_{\epsilon\epsilon\epsilon'}$ is given in equation (3.30). The one-point functions $\langle(\epsilon'^{TIM})_A\rangle_{RG}$ and $\langle(1)_A\rangle_{RG}$ are in [1] equation (2.56), giving $\frac{\langle(\epsilon'^{TIM})_A\rangle_{RG}}{\langle(1)_A\rangle_{RG}} = -\frac{1}{2}\left(\frac{\frac{5}{8}+\frac{\sqrt{5}}{8}}{\frac{5}{8}-\frac{\sqrt{5}}{8}}\right)^{1/4}$, in agreement with the coefficient of $z^{3/5}$ above. The coefficient of $z^2$ follows from the OPE coefficient and one-point function for the stress tensor. It can easily be calculated using the usual Virasoro conformal blocks, where the coefficient of $z^2$ is $\frac{2h^2}{c}$. In this case, there is an additional suppression for the

one-point function of $T$ in the presence of the RG brane, and this suppression can be read off from (A.19) where we see that $T^{TIM}$s overlap with itself reflected across the RG brane picks up a factor of $\frac{3}{k(k+2)} = \frac{3}{8}$. Therefore, the prediction for the coefficient of $z^2$ is $\frac{3}{8}\frac{2h^2}{c} = \frac{3}{280}$ since $h = h_\epsilon^{TIM} = \frac{1}{10}$ and $c = c^{TIM} = \frac{7}{10}$. Again this agrees with the coefficient of $z^2$ above. Finally, we can fix one more coefficient by using the long-distance limit $\langle\epsilon(x_1,\overline{x}_1)\epsilon(x_2,\overline{x}_2)\rangle_{RG} \sim \langle\epsilon(x_1,\overline{x}_1)\rangle_{RG}\langle\epsilon(x_2,\overline{x}_2)\rangle_{RG} = 0$ at $x_1 - x_2 \to \infty$. And indeed, there is a nontrivial cancellation among the four conformal blocks in $G^{TIM}$ so that the leading large $z$ term vanishes. This condition completes the independent derivation of all the coefficients in $G^{TIM}$.

To get the correlator at finite volume, we can conformally map the correlator from the upper half plane (UHP) to the strip. Without loss of generality, we can choose units where the volume is $2\pi$, and reintroduce an arbitrary volume later by scaling. The conformal mapping from the UHP to the strip is ($t_E$ being Euclidean time)

$$x = e^{t_E + i\theta}, \tag{3.15}$$

so that the RG brane located at $\mathrm{Im}(x) = 0$ gets mapped to $\theta = 0, \pi$. In the folded description, both TIM and Ising are in the UHP, and get mapped to the range $0 < \theta < \pi$, whereas in the unfolded description, TIM is in, say, the UHP whereas Ising is in the lower half-plane (LHP), so that they get mapped to $0 < \theta < \pi$ and $\pi < \theta < 2\pi$, respectively. When we set up our DMRG calculation, we will choose periodic boundary conditions, in order to match the unfolded description in finite volume. It is straightforward to evaluate the CFT expression for the correlator at arbitrary times, but we will focus on the case of equal-time correlators and set $t_E = 0$. Then, in terms of the function $G_{\epsilon\epsilon}^{TIM}(z)$ from (3.9), the finite-volume two-point function of $\epsilon^{TIM}$ in the presence of the RG brane is

$$\langle\epsilon^{TIM}(\theta_1)\epsilon^{TIM}(\theta_2)\rangle_{RG} \propto \left(\frac{1}{\sin^2\left(\frac{\theta_1-\theta_2}{2}\right)}\right)^{\frac{1}{5}} G_{\epsilon\epsilon}^{TIM}\left(\frac{\sin^2\left(\frac{\theta_1-\theta_2}{2}\right)}{\sin^2\left(\frac{\theta_1+\theta_2}{2}\right)}\right). \tag{3.16}$$

The proportionality constant in the above equation depends on the normalization of $\epsilon^{TIM}$ and can easily be fixed by matching the OPE singularity at $\theta_1 \sim \theta_2$.

### 3.3 $\langle\epsilon^{IM}\epsilon^{IM}\rangle_{RG}$

The calculation of the two-point function of the $\epsilon$ operator on the Ising model side of the RG brane closely follows the derivation of that on the TIM side in the previous subsection. The difference is that $\epsilon^{IM}$ maps in the $\widetilde{\mathcal{B}}$ theory to the operator $(\sigma^{IM}\sigma'^{TIM})_B$, and so we need to use the TIM $\sigma'$ OPE coefficients and conformal blocks. The TIM $\sigma' \times \sigma'$ OPE only contains two primary operators, $1$ and $\epsilon''$. Because $\sigma'$ has a null descendant at level 2, its conformal blocks can easily be calculated by standard methods. They are

$$G_{\sigma'\sigma'1}^{TIM}(z) = \frac{(1-z)^{5/8}\,{}_2F_1\left(-\frac{1}{4},\frac{5}{4};-\frac{1}{2};z\right)}{z^{7/8}}, \quad G_{\sigma'\sigma'\epsilon''}^{TIM}(z) = (1-z)^{5/8}z^{5/8}\,{}_2F_1\left(\frac{5}{4},\frac{11}{4};\frac{5}{2};z\right). \tag{3.17}$$

The corresponding OPE coefficients are

$$C_{\sigma'\sigma'1}^{TIM} = 1, \quad C_{\sigma'\sigma'\epsilon''}^{TIM} = \frac{7}{8}. \tag{3.18}$$

As before, the full correlator is a sum over these conformal blocks with coefficients given by the bulk OPE coefficients times the RG brane one-point functions:

$$\langle(\sigma^{IM}\sigma'^{TIM})(x_1,\overline{x}_1)(\sigma^{IM}\sigma'^{TIM})(x_2,\overline{x}_2)\rangle_{RG} = \frac{1}{|x_1-x_2|^2}G_{\epsilon\epsilon}^{IM}\left(\left|\frac{x_1-x_2}{x_1-\overline{x}_2}\right|^2\right),$$

$$G_{\epsilon\epsilon}^{IM}(z) \equiv z \sum_{\substack{\phi_i=1,\epsilon \\ \phi_j=1,\epsilon''}}' \langle(\phi_i^{IM}\phi_j^{TIM})_B|RG\rangle C_{\sigma\sigma\phi_i}^{IM}C_{\sigma'\sigma'\phi_j}^{TIM}G_{\sigma\sigma\phi_i}^{IM}(z)G_{\sigma'\sigma'\phi_j}^{TIM}(z). \tag{3.19}$$

The sum on $\phi_i, \phi_j$ includes only two terms in total, the combination $(\phi_i = 1, \phi_j = 1)$ and $(\phi_i = \epsilon, \phi_j = \epsilon')$. The one-point functions were given explicitly in (3.13). The finite-volume, equal-time correlator is

$$\langle\epsilon^{IM}(\theta_1)\epsilon^{IM}(\theta_2)\rangle_{RG} \propto \frac{1}{\sin^2\left(\frac{\theta_1-\theta_2}{2}\right)}G_{\epsilon\epsilon}^{IM}\left(\frac{\sin^2\left(\frac{\theta_1-\theta_2}{2}\right)}{\sin^2\left(\frac{\theta_1+\theta_2}{2}\right)}\right). \tag{3.20}$$

### 3.4 Branching Rule and Two-point Function for $\epsilon'^{TIM}$

Finally, let us work out the CFT prediction for a slightly more complicated example, that of the RG brane two-point function of $\epsilon'^{TIM}$. Now, the operator maps to a linear combination of primaries in the $\widetilde{\mathcal{B}}$ theory, which we can work out using the action of the chiral algebra in the $\mathcal{A}$ and $\widetilde{\mathcal{B}}$ descriptions. The branching rule tells us that $1^{IM}\epsilon'^{TIM}(x)_A = 1^{IM}\epsilon'^{TIM}(x)_B \oplus \epsilon^{IM}\epsilon^{TIM}(x)_B + 1^{IM}\epsilon'^{TIM}(x)_B \oplus \epsilon^{IM}\epsilon^{TIM}(x)_B$. Our task now is to determine the precise decomposition coefficients $a$ and $b$ for

$$1^{IM}\epsilon'^{TIM}(x)_A = a1^{IM}\epsilon'^{TIM}(x)_B + b\epsilon^{IM}\epsilon^{TIM}(x)_B. \tag{3.21}$$

This can be done by using the first equation in Equ. (A.19), which tells us that

$$L_{0;A}^{IM} = \frac{5}{8}L_{0;B}^{TIM} + \frac{\sqrt{15}}{8}(G\psi)_0 + \frac{1}{8}L_{0;B}^{IM}, \tag{3.22}$$

together with

$$L_{0;A}^{IM}1^{IM}\epsilon'(x)_A = 0. \tag{3.23}$$

We have

$$0 = \left(\frac{3a + \sqrt{15}b\sqrt{2h_{\epsilon^{TIM}}}}{8}1^{IM} \otimes \epsilon'^{TIM} + \frac{5\sqrt{15}\sqrt{2h_{\epsilon^{TIM}}}a + 5b}{40}\epsilon^{IM} \otimes \epsilon^{TIM}\right), \tag{3.24}$$

and so

$$\sqrt{3}a + b = 0. \tag{3.25}$$

Hence we have the decomposition up to an overall constant.

Translating to the B-theory, the correlator that we are interested in is

$$\left\langle \left(1^{IM}(x_1)\epsilon'^{TIM}(x_1) - \sqrt{3}\epsilon^{IM}(x_1)\epsilon^{TIM}(x_1)\right)\left(1^{IM}(x_2)\epsilon'^{TIM}(x_2) - \sqrt{3}\epsilon^{IM}(x_2)\epsilon^{TIM}(x_2)\right)\right\rangle$$

$$=\langle 1^{IM}(x_1)\epsilon'^{TIM}(x_1)1^{IM}(x_2)\epsilon'^{TIM}(x_2)\rangle + 3\langle\epsilon^{IM}(x_1)\epsilon^{TIM}(x_1)\epsilon^{IM}(x_2)\epsilon^{TIM}(x_2)\rangle$$

$$- \sqrt{3}\langle 1^{IM}(x_1)\epsilon'^{TIM}(x_1)\epsilon^{IM}(x_2)\epsilon^{TIM}(x_2)\rangle - \sqrt{3}\langle\epsilon^{IM}(x_1)\epsilon^{TIM}(x_1)1^{IM}(x_2)\epsilon'^{TIM}(x_2)\rangle,$$

$$(3.26)$$

and we will compute the resulting four correlators separately.

To compute these correlators, we follow the same procedure as for Equ. (3.6) doing the operator production expansion (OPE) in the bulk first and computing the resulting one-point functions (as indicated in Fig.2). The OPE is given by the (tensor product of the) OPE from the Ising model and the Tricritical Ising Model. The conformal blocks we get are those for chiral four point functions in the Ising model and tricitical Ising model. These bulk OPE expansions can be read from

$$[\epsilon^{IM}] \times [\epsilon^{IM}] = [1^{IM}], \quad [\epsilon^{TIM}] \times [\epsilon^{TIM}] = [1^{TIM}] + [\epsilon^{TIM}],$$

$$[\epsilon'^{TIM}] \times [\epsilon'^{TIM}] = [1^{TIM}] + [\epsilon'^{TIM}], \quad [\epsilon^{TIM}] \times [\epsilon'^{TIM}] = [\epsilon^{TIM}] + [\epsilon''^{TIM}].$$

$$(3.27)$$

The results are given by

$$\langle 1^{IM}(x_1)\epsilon'^{TIM}(x_1)1^{IM}(x_2)\epsilon'^{TIM}(x_2)\rangle$$

$$=\frac{z^{\frac{2}{5}}(1-z)^{\frac{2}{5}}\left(H_{11}^{I}(z) + H_{1\epsilon'}^{I}(z)\right)}{(x_1-x_2)^{\frac{1}{15}}(x_1-\overline{x}_1)^{\frac{1}{15}}(x_1-\overline{x}_1)^{\frac{1}{15}}(x_2-\overline{x}_1)^{\frac{1}{15}}(x_1-\overline{x}_2)^{\frac{1}{15}}(x_2-\overline{x}_2)^{\frac{1}{15}}(x_2-\overline{x}_1)^{\frac{1}{15}}(\overline{x}_1-\overline{x}_2)^{\frac{1}{15}}},$$

$$\langle\epsilon^{IM}(x_1)\epsilon^{TIM}(x_1)\epsilon^{IM}(x_2)\epsilon^{TIM}(x_2)\rangle$$

$$=\frac{z^{\frac{2}{5}}(1-z)^{\frac{2}{5}}\left(H_{11}^{II}(z) + H_{1\epsilon'}^{II}(z)\right)}{(x_1-x_2)^{\frac{1}{15}}(x_1-\overline{x}_1)^{\frac{1}{15}}(x_1-\overline{x}_1)^{\frac{1}{15}}(x_2-\overline{x}_1)^{\frac{1}{15}}(x_1-\overline{x}_2)^{\frac{1}{15}}(x_2-\overline{x}_2)^{\frac{1}{15}}(x_2-\overline{x}_1)^{\frac{1}{15}}(\overline{x}_1-\overline{x}_2)^{\frac{1}{15}}},$$

$$\langle 1^{IM}(x_1)\epsilon'^{TIM}(x_1)\epsilon^{IM}(x_2)\epsilon^{TIM}(x_2)\rangle$$

$$=\frac{z^{\frac{2}{5}}(1-z)^{\frac{2}{5}}\left(H_{\epsilon\epsilon}^{III}(z) + H_{\epsilon\epsilon''}^{III}(z)\right)}{(x_1-x_2)^{\frac{1}{15}}(x_1-\overline{x}_1)^{\frac{1}{15}}(x_1-\overline{x}_1)^{\frac{1}{15}}(x_2-\overline{x}_1)^{\frac{1}{15}}(x_1-\overline{x}_2)^{\frac{1}{15}}(x_2-\overline{x}_2)^{\frac{1}{15}}(x_2-\overline{x}_1)^{\frac{1}{15}}(\overline{x}_1-\overline{x}_2)^{\frac{1}{15}}},$$

$$\langle\epsilon^{IM}(x_1)\epsilon^{TIM}(x_1)1^{IM}(x_2)\epsilon'^{TIM}(x_2)\rangle$$

$$=\frac{z^{\frac{2}{5}}(1-z)^{\frac{2}{5}}\left(H_{\epsilon\epsilon}^{IV}(z) + H_{\epsilon\epsilon''}^{IV}(z)\right)}{(x_1-x_2)^{\frac{1}{15}}(x_1-\overline{x}_1)^{\frac{1}{15}}(x_1-\overline{x}_1)^{\frac{1}{15}}(x_2-\overline{x}_1)^{\frac{1}{15}}(x_1-\overline{x}_2)^{\frac{1}{15}}(x_2-\overline{x}_2)^{\frac{1}{15}}(x_2-\overline{x}_1)^{\frac{1}{15}}(\overline{x}_1-\overline{x}_2)^{\frac{1}{15}}},$$

$$(3.28)$$

where the relevant conformal blocks are

$$H_{11}^{I}(z) = \langle (1^{IM} 1^{TIM})_B | RG \rangle \rangle C_{\epsilon'\epsilon'1}^{TIM} \frac{(z^2 - z + 1) \, _2F_1\left(-\frac{2}{5}, \frac{1}{5}; \frac{2}{5}; z\right)}{z^{6/5}(1-z)^{6/5}} \,,$$

$$H_{1\epsilon'}^{I}(z) = \langle (1^{IM} \epsilon'^{TIM})_B | RG \rangle \rangle C_{\epsilon'\epsilon'\epsilon'}^{TIM} \frac{(z^2 - z + 1) \, _2F_1\left(\frac{1}{5}, \frac{4}{5}; \frac{8}{5}; z\right)}{z^{3/5}(1-z)^{6/5}} \,,$$

$$H_{11}^{II}(z) = \langle (1^{IM} 1^{TIM})_B | RG \rangle \rangle C_{\epsilon\epsilon1}^{IM} C_{\epsilon\epsilon1}^{TIM} \frac{1 - z + z^2}{z(1-z)} \frac{_2F_1\left(-\frac{2}{5}, \frac{1}{5}; \frac{2}{5}; z\right)}{\sqrt[5]{z}\sqrt[5]{1-z}} \,,$$

$$H_{1\epsilon'}^{II}(z) = \langle (1^{IM} \epsilon'^{TIM})_B | RG \rangle \rangle C_{\epsilon\epsilon1}^{IM} C_{\epsilon\epsilon\epsilon'}^{TIM} \frac{1 - z + z^2}{z(1-z)} \frac{_2F_1\left(\frac{1}{5}, \frac{4}{5}; \frac{8}{5}; z\right)}{z^{-\frac{2}{5}}\sqrt[5]{1-z}} \,,$$

$$H_{\epsilon\epsilon}^{III}(z) = \langle (\epsilon^{IM} \epsilon^{TIM})_B | RG \rangle \rangle C_{\epsilon'\epsilon\epsilon}^{TIM} \frac{1}{1-z} \frac{_2F_1\left(-\frac{6}{5}, \frac{1}{5}; -\frac{2}{5}; z\right)}{\sqrt[5]{1-z}z^{3/5}} \,,$$

$$H_{\epsilon\epsilon''}^{III}(z) = \langle (\epsilon^{IM} \epsilon''^{TIM})_B | RG \rangle \rangle C_{\epsilon'\epsilon\epsilon''}^{TIM} \frac{1}{1-z} \frac{z^{4/5} \, _2F_1\left(\frac{1}{5}, \frac{8}{5}; \frac{12}{5}; z\right)}{\sqrt[5]{1-z}} \,,$$

$$H_{\epsilon\epsilon}^{IV}(z) = \langle (\epsilon^{IM} \epsilon^{TIM})_B | RG \rangle \rangle C_{\epsilon'\epsilon\epsilon}^{TIM} \frac{1}{1-z} \frac{_2F_1\left(-\frac{6}{5}, \frac{1}{5}; -\frac{2}{5}; z\right)}{\sqrt[5]{1-z}z^{3/5}} \,,$$

$$H_{\epsilon\epsilon''}^{IV}(z) = \langle (\epsilon^{IM} \epsilon''^{TIM})_B | RG \rangle \rangle C_{\epsilon'\epsilon\epsilon''}^{TIM} \frac{1}{1-z} \frac{z^{4/5} \, _2F_1\left(\frac{1}{5}, \frac{8}{5}; \frac{12}{5}; z\right)}{\sqrt[5]{1-z}} \,.$$

(3.29)

The OPE coefficients are

$$C_{\epsilon\epsilon1}^{IM} = 1 \,,$$

$$C_{\epsilon'\epsilon'1}^{TIM} = 1 \,,$$

$$C_{\epsilon'\epsilon'\epsilon'}^{TIM} = \frac{\pi^{5/4}\Gamma\left(\frac{2}{5}\right)\Gamma\left(\frac{11}{5}\right)}{\sqrt[5]{2}\Gamma\left(\frac{9}{10}\right)\left(\Gamma\left(\frac{1}{5}\right)\Gamma\left(\frac{13}{10}\right)\right)^{3/2}\sqrt{3\Gamma\left(\frac{7}{5}\right)}} \,,$$

$$C_{\epsilon\epsilon1}^{TIM} = 1 \,,$$

$$C_{\epsilon\epsilon\epsilon'}^{TIM} = \frac{\sqrt{\frac{\Gamma\left(-\frac{3}{10}\right)\Gamma\left(\frac{2}{5}\right)\Gamma\left(\frac{7}{5}\right)}{\Gamma\left(-\frac{2}{5}\right)\Gamma\left(\frac{3}{10}\right)\Gamma\left(\frac{8}{5}\right)}}}{2^{3/5}} \,,$$

$$C_{\epsilon'\epsilon\epsilon}^{TIM} = \frac{\sqrt{\frac{\Gamma\left(-\frac{3}{10}\right)\Gamma\left(\frac{2}{5}\right)\Gamma\left(\frac{7}{5}\right)}{\Gamma\left(-\frac{2}{5}\right)\Gamma\left(\frac{3}{10}\right)\Gamma\left(\frac{8}{5}\right)}}}{2^{3/5}} \,,$$

$$C_{\epsilon'\epsilon\epsilon''}^{TIM} = \frac{3}{7} \,,$$

(3.30)

and the boundary operator expansion coefficients are

$$\langle (1^{IM} 1^{TIM})_B | RG \rangle\rangle = \frac{1}{2} \sqrt{S_{0,0}^{(1)} S_{0,0}^{(3)}} ,$$

$$\langle (1^{IM} \epsilon'^{TIM})_B | RG \rangle\rangle = \frac{1}{2} \sqrt{S_{0,0}^{(1)} S_{0,2}^{(3)}} ,$$

$$\langle (\epsilon^{IM} \epsilon^{TIM})_B | RG \rangle\rangle = -\frac{1}{2} \sqrt{S_{0,0}^{(1)} S_{0,2}^{(3)}} ,$$

$$\langle (\epsilon^{IM} \epsilon''^{TIM})_B | RG \rangle\rangle = -\frac{1}{2} \sqrt{S_{0,0}^{(1)} S_{0,0}^{(3)}} .$$

(3.31)

## 4  Comparison of Lattice and CFT Calculations

In this section, we numerically study the RG brane system using a DMRG analysis. Following [3], we apply DMRG to the Hamiltonian (2.2). Recall that for (2.2) when $h = 2$ the system is in the IM phase whereas for $h = 1.62$ the system is in the TIM phase, and the RG brane system is realized on the lattice by tuning $h = 2$ on half the lattice sites and $h = 1.62$ on the other half, with periodic boundary conditions taken at the endpoints.

As mentioned in Sec. 2.2, the DMRG algorithm is controlled by a few key parameters. For the results below we use a lattice with $N = 180$ distinct sites (540 total sites, since each physical site requires a separate site for $\mu_{i,a}, \mu_{i,b}$ and $\sigma_i$) unless otherwise noted. First we will find the map between lattice operators and CFT operators. Then, we numerically compute one-point and two-point functions in DMRG, and compare them with the analytic CFT predictions from Sec. 3.

### 4.1  Mapping of Operators

Because the lattice is a microscopic structure versus the continuum field theory that we aim to compare it to, we need to verify what lattice operators correspond to what CFT operators. We do this by expanding the lattice operators in terms of the CFT operators and then determining the expansion coefficients. In our case, the $\mu$ lattice operator (i.e. $\mu_{i,b}^z$ and $\mu_{i,a}^z$) in the Ising domain can be expanded as

$$\mu^z = A_\epsilon^{(I)} \, \epsilon + A_{\partial^2 \epsilon}^{(I)} \, \partial^2 \epsilon + \cdots$$

(4.1)

where $\epsilon$ is the Ising CFT energy density operator, the $A$s are the expansion coefficients and the dots are higher decedent operators. In the tricritical Ising domain the expansion is,

$$\mu^z = A_\epsilon^{(T)} \, \epsilon + A_{\epsilon'}^{(T)} \, \epsilon' + A_{\partial^2 \epsilon}^{(T)} \, \partial^2 \epsilon + \cdots$$

(4.2)

where $\epsilon$, $\epsilon'$, etc., are the tricritical CFT energy density operators. The fact that the lattice $\mu$ operator expands to energy operators can be understood as follows. Adding $\sum_i \mu_{i,b}^z$ (or $\sum_i \mu_{i,a}^z$) to the Hamiltonian Equ. (2.2) breaks the $\mathbb{Z}_2$ symmetry which acts on the $\mu$-sector

as sending $\mu_{i,a}^z$ to $-\mu_{i,b}^z$ and $\mu_{i,b}^z$ to $-\mu_{i+1,a}^z$ and on the $\sigma$-sector as the high-temperature/low-temperature duality transform $\sigma_i^z \sigma_{i+1}^z \leftrightarrow \sigma_i^x$. Hence when are in the Ising phase this deformation breaks the high-temperature/low-temperature duality symmetry[11] which maps $\epsilon$ to $-\epsilon$ so this is equivalent at leading order to an $\epsilon$ deformation. On the other hand, when we are at the critical point ($h = h_*$) the low-energy theory is TIM and such a deformation would gap out the system. Hence this is a relevant deformation. However, in the TIM case a $\mathbb{Z}_2$ symmetry we have is

$$\epsilon^{TIM} \to -\epsilon^{TIM}, \quad \epsilon'^{TIM} \to \epsilon'^{TIM}, \quad \text{and } \epsilon''^{TIM} \to -\epsilon''^{TIM}, \tag{4.3}$$

which suggests that the expansion of the lattice $\mu$ operator is given by the $\epsilon^{TIM}, \epsilon''^{TIM}$ and their descendants.

To determine the coefficients we use DMRG to calculate finite-volume two-point functions on the lattice in the IM and TIM phases. The equal-time finite-volume two-point function of a primary operator $\mathcal{O}$ is given by

$$\langle \mathcal{O}(x_1)\mathcal{O}(x_2) \rangle = \frac{(\frac{\pi}{L})^{2\Delta_\mathcal{O}}}{\sin\left(\frac{\pi x_{12}}{L}\right)^{2\Delta_\mathcal{O}}} \sim \frac{1}{x_{12}^{2\Delta_\mathcal{O}}}, \tag{4.4}$$

where we have chosen the canonical normalization of $\mathcal{O}$ so that its short-distance OPE singularity is $x_{12}^{-2\Delta_\mathcal{O}}$, as shown. Therefore, for the case of (4.1), we fit numeric data to a function of the form

$$\langle \mu(x_1)\mu(x_2) \rangle_{\text{TIM}} \approx (A_\epsilon^{(T)})^2 \frac{(\frac{\pi}{L})^{2\Delta_\epsilon}}{\sin\left(\frac{\pi x_{12}}{L}\right)^{2\Delta_\epsilon}}. \tag{4.5}$$

with $\Delta_\epsilon = \frac{1}{5}$, and similarly with the Ising case using $\Delta_\epsilon = 1$. Doing this we find, using a $N = 180$ site lattice,

$$A_\epsilon^{(I)} = 0.81, \qquad A_\epsilon^{(T)} = 0.61. \tag{4.6}$$

In Fig. 3, we show a comparison of the form (4.5) to the result of our DMRG computation for the $\langle \mu_i \mu_j \rangle$ two-point function on the TIM and Ising side, verifying a similar comparison in [3].

It is more difficult to obtain very precise results for the coefficients of the subleading operators in (4.1) and (4.2). However, such contributions should be parameterically suppressed by $|i - j|^2$ compared to the leading contribution at long distances. Empirically, we find that this suppression is roughly a factor of

$$\frac{b}{|i - j|^2} \tag{4.7}$$

---

[11]In the free fermion representation this is the time-reversal symmetry which prevents the system from being gapped.

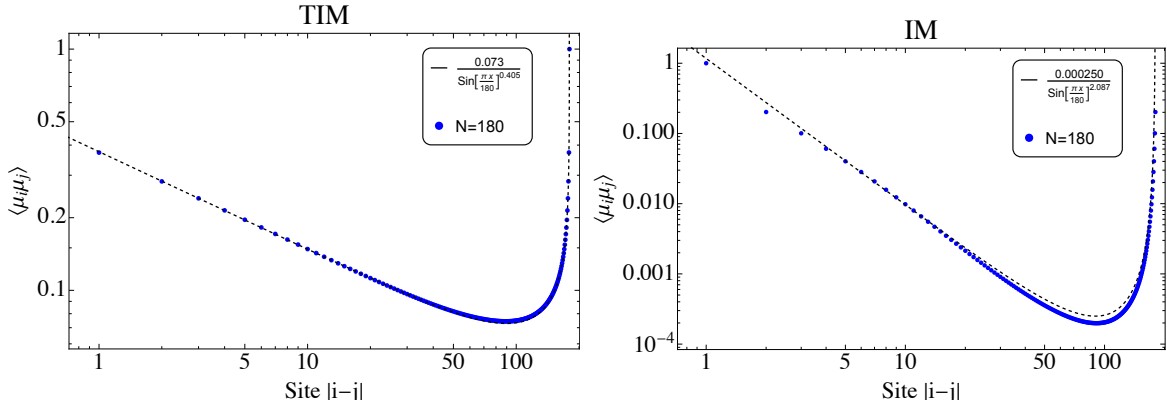

**Figure 3**. DMRG computation of the $\langle \mu_i \mu_j \rangle$ two-point function for the lattice model Equ. (2.2) in the Tricritical Ising ($h = 1.62$) and Ising ($h = 2$) phases.

(relative to the leading contribution) with $b \sim 20$ for Ising and $b \sim 1$ for TIM.[12] So we see that the lattice $\mu$ operators flow to a linear combination of the energy density operators, with the dominant CFT operator being the $\epsilon$ energy density in the respective phase, as expected. Moreover, lattice corrections from subleading operators in (4.1) and (4.2) should be small for $|i - j| \gtrsim 5$ for Ising and $|i - j| \gtrsim 1$ for TIM. Indeed this is consistent with what we will see in Fig. 4 when we compare the CFT and lattice calculations of the $\mu_i$ two-point function in the presence of the RG brane.

## 4.2 Numeric Comparisons

We start by using DMRG to compute one-point functions. In [1] one-point functions were verified using conformal perturbation theory for large $k$ Virasoro minimal CFTs; using DMRG allows us to check these formulas at finite $k$ (in our case $k = 1$).

To compute the one-point function, we fit the DMRG data of the two-point function $\langle \mu_{j_*+x} \mu_{j_*-x} \rangle_{RG}$, where $j_*$ is the location of the RG brane, to the form

$$\langle \mu_{j_*-x} \mu_{j_*+x} \rangle_{RG} = A_\epsilon^{(I)} A_\epsilon^{(T)} \langle \epsilon^{TIM}(x) \epsilon^{IM}(-x) \rangle_{RG} = A_\epsilon^{(I)} A_\epsilon^{(T)} B \left( \frac{\frac{\pi}{L}}{\sin\left(\frac{2x\pi}{L}\right)} \right)^\Delta, \qquad (4.8)$$

where $\Delta = \Delta_\epsilon^{IM} + \Delta_\epsilon^{TIM} = 1.2$ and $A_\epsilon$s are taken from the fits in the previous subsection. As usual, we have performed a conformal map to take into account finite volume effects. We obtain the result

$$B = 0.64 \pm 0.01, \qquad (4.9)$$

---

[12]A possible explanation for the smallness of this value of $b$ in TIM is as follows. We expect that the coefficient $A_{\epsilon'}$ actually vanishes due to the generalization of Kramers-Wannier symmetry to TIM (i.e., the $\widehat{N}$ Verlinde line [22]), so the only contribution to $b$ comes from $\partial^2 \epsilon$ in both Ising and in TIM. Then, the cross-terms $\langle \epsilon \partial^2 \epsilon \rangle$ are parametrically suppressed relative to the leading terms $\langle \epsilon \epsilon \rangle$ by $\frac{\langle \epsilon \partial^2 \epsilon \rangle}{\langle \epsilon \epsilon \rangle} \sim \Delta_\epsilon^2$, which is 1 in Ising and 1/25 in TIM.

where we have estimated the error by performing the fit over different ranges of lattice points.

Now we want to compare this numerical result with the prediction from the RG brane construction. The $\mathcal{A}$-theory operator $\epsilon^{IM}(x)\epsilon^{TIM}(x)$ can be written as $\phi_{r,s}^{(k+1)}\phi_{s,t}^{(k+2)}$ where $k = 2, r = 2, s = 1$ and $t = 2$. This tells us that $d = 2, \widetilde{d} = 2$ and $d' = 1$ or $3$. Hence this $\mathcal{A}$-theory operator translates into the $\widetilde{\mathcal{B}}$-theory as the linearly combination of the operators $[2, 1; 2] \otimes [2, 2; 1] = \epsilon^{TIM}\epsilon^{IM}(x)$ and $[2, 3; 2] \otimes [2, 2; 3] = G_{-\frac{1}{2}}\overline{G}_{-\frac{1}{2}}\epsilon^{TIM}1^{IM}(x) = \epsilon'^{TIM}1^{IM}(x)$.[13] Let's write

$$\epsilon^{IM}\epsilon^{TIM}(x)_A = \alpha\epsilon^{TIM}\epsilon^{IM}(x)_B + \beta G_{-\frac{1}{2}}\epsilon^{TIM}1^{IM}(x)\,, \tag{4.10}$$

where we only focus on the holomorphic part and the normalization of the operator gives the constraint

$$1 = \alpha^2 + 2\beta^2 h_{\epsilon^{TIM}} = \alpha^2 + \frac{1}{5}\beta^2\,. \tag{4.11}$$

We can find the coefficients $\alpha$ and $\beta$ as in Equ. (3.21) using the following relation

$$L_{0;A}^{IM} = \frac{5}{8}L_{0;B}^{TIM} + \frac{\sqrt{15}}{8}(G\psi)_0 + \frac{1}{8}L_{0;B}^{IM}\,, \tag{4.12}$$

which tells us that

$$\frac{\sqrt{15}}{40}\beta = \frac{3}{8}\alpha\,, \quad \frac{1}{8}\beta = \frac{\sqrt{15}}{8}C_{\epsilon\epsilon1}^{IM}\alpha\,. \tag{4.13}$$

The fact that $C_{\epsilon\epsilon1}^{IM} = 1$ tells us that this two equations are consistent. Then combining with Equ. (4.11) we have

$$\alpha = \sqrt{\frac{5}{8}}\,, \beta = 5\sqrt{\frac{3}{8}}\,. \tag{4.14}$$

As a result, we have

$$\langle\langle(\epsilon^{IM}\epsilon^{TIM})_A|RG\rangle\rangle = (-\alpha^2 + 2h_{\epsilon^{TIM}}\beta^2)\frac{\sqrt{S_{0,1}^{(1)}S_{0,1}^{(3)}}}{S_{0,0}^{(2)}} = \frac{1}{2}\frac{\sqrt{S_{0,1}^{(1)}S_{0,1}^{(3)}}}{S_{0,0}^{(2)}}\,, \tag{4.15}$$

where the minus sign in the first step comes from the fact that the RG brane maps $\psi$ (i.e. the homolorphic part of $\epsilon_B^{IM}$) to $-\overline{\psi}$ (i.e. the antiholomorphic part of $\epsilon_B^{IM}$). Moreover, there is one more step before we can match the numerical result Equ. (4.9). We have to take into account of the fact that in numerics the identity operator is normalized to one and so we have to consider instead

$$\frac{\langle\langle(\epsilon^{IM}\epsilon^{TIM})_A|RG\rangle\rangle}{\langle\langle(1^{IM}1^{TIM})_A|RG\rangle\rangle} = \frac{1}{2}\frac{\sqrt{S_{0,1}^{(1)}S_{0,1}^{(3)}}}{S_{0,0}^{(2)}}\frac{S_{0,0}^{(2)}}{\sqrt{S_{0,0}^{(1)}S_{0,0}^{(2)}}} = \frac{1}{2}(\frac{5+\sqrt{5}}{5-\sqrt{5}})^{\frac{1}{4}} \sim 0.63601\,, \tag{4.16}$$

which perfectly matches the numerical result Equ. (4.9).

---

[13]Here we used the fact that the $\widetilde{\mathcal{B}}$-theory is the supersymmetric representation of the Tricritical Ising Model for our case $k = 2$. $G_{-\frac{1}{2}}$ and $\overline{G}_{-\frac{1}{2}}$ are the first creation operators of the supersymmetry generators $G$ and $\overline{G}$ in the NS sector.

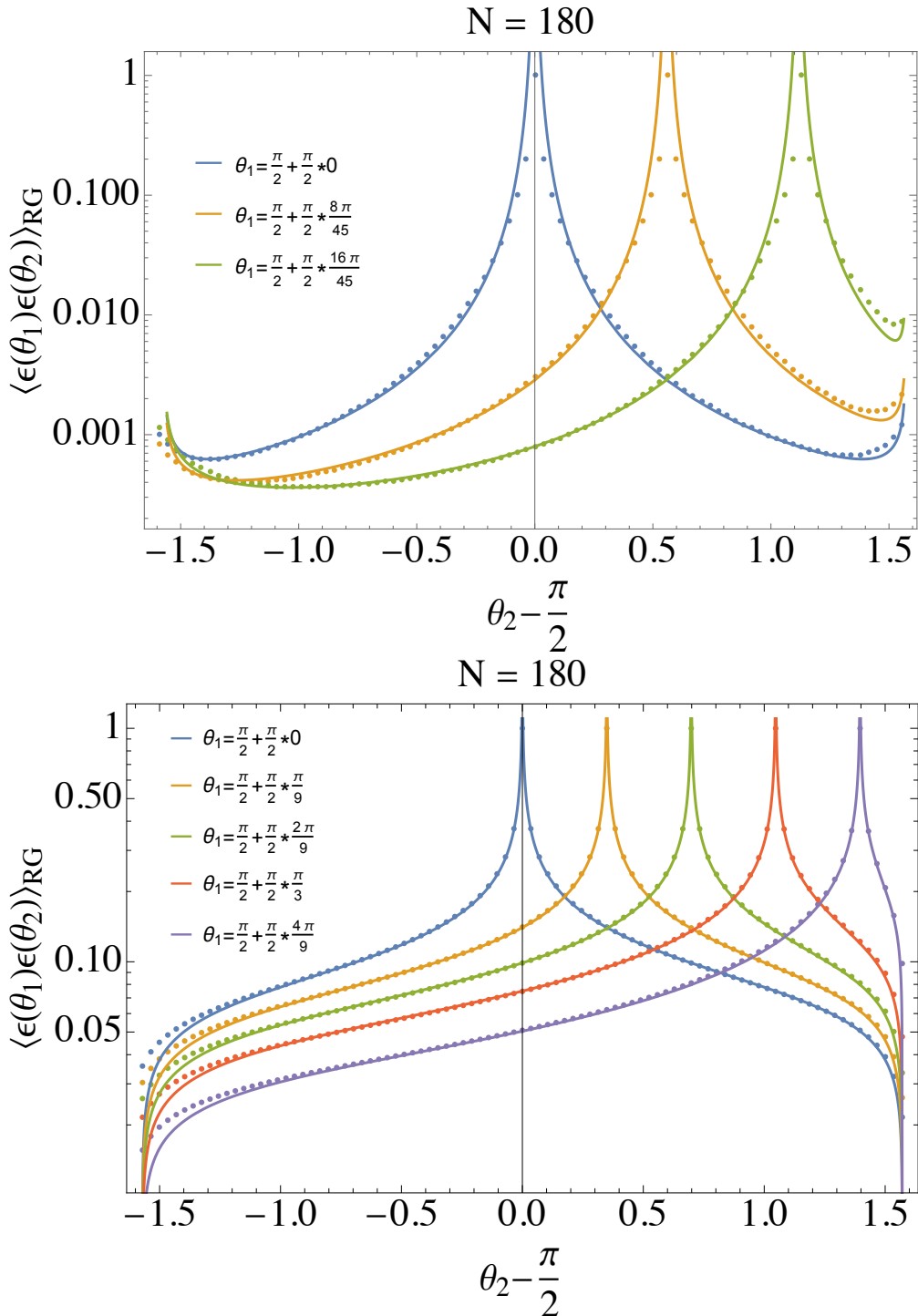

**Figure 4**. Comparison of $\langle\mu_i\mu_j\rangle$ in the RG brane lattice description vs. the equivalent CFT result Equ. (3.9). *Top:* The IM side. *Bottom:* The TIM side.

Finally, we can compare our CFT result Equ. (3.9) for $\langle\epsilon\epsilon\rangle_{\mathrm{RG}}$ in the presence of the RG brane to the DMRG calculation. Fig. 4 shows the result of this comparison for the computing $\langle\mu_i\mu_j\rangle$ on the Ising and Tricritical Ising sides of the RG brane, respectively, and shows remarkably good agreement between the CFT and DMRG calculations. This is one of the main results of the paper, and provides a highly nontrivial check of our methods for computing correlators in RG brane backgrounds.

## 5    Experimental Realizations

In this section, we comment on potential experimental realizations of the RG domain wall we studied in this paper, based on some experimental proposals in the literature for obtaining the Tricritical Ising Model fixed point. There are two types of systems that we will focus on and they each have their own pros and cons. The first type of systems we will consider is (2+1)-dimensional Type IIID topological superconductors [2, 3, 23] and the second type of systems we are interested in is the (1+1)-dimensional Rydberg chain [24].

### 5.1    Engineering the RG Domain Wall on the Boundary of a Topological Super-conductor

Let's consider the edge of a (2+1)-dimensional Type III D topological superconductor with the bulk in the gapped topological phase (a discussion of the physics in a toy model for such systems is given in Appendix.C).[14] For such a system we have two Majorana modes $\chi_L$ and $\chi_R$ on the edge whose gap is protected by a time-reversal symmetry. The Lagrangian is given by

$$L = \frac{i}{2}\int dx\Big[\chi_R(x,t)(\partial_t + \partial_x)\chi_R(x,t) + \chi_L(x,t)(\partial_t - \partial_x)\chi_L(x,t)\Big], \qquad (5.1)$$

which is the (1+1)-d Ising Model conformal field theory. The two Majorana fermions $\chi_R$ and $\chi_L$ are respectively of spin down and spin up and the time-reversal symmetry exchanges these two Majorana fermions and meanwhile it protects the system from being gapped (see Appendix.C). Hence the spontaneous breaking of the time-reversal symmetry (or the magnetic ordering) provides a portal to realize a different phase (the trivial gapped phase) and the critical point of the phase transition between this new gapped phase the gapless phase Equ. (5.1) provides a chance to realize a different CFT. Such a phase transition can be characterized by an order parameter $\phi(x,t)$ which transforms under the time-reversal symmetry by a minus sign. The dynamics of such an order parameter is universally controlled by the

---

[14]We thank Tarun Grover for discussions of this approach, and suggestions for how to implement the RG brane in this context.

usual $\phi^4$-model. The full Lagrangian of the system is given by

$$
\begin{aligned}
L_{tot} =& \frac{i}{2} \int dx \Big[ \chi_R(x,t)(\partial_t + \partial_x)\chi_R(x,t) + \chi_L(x,t)(\partial_t - \partial_x)\chi_L(x,t) \Big] \\
&+ \frac{1}{2} \int dx \Big[ (\partial_t \phi)^2(x,t) - (\partial_x \phi)^2(x,t) + m^2 \phi^2(x,t) - u\phi^4(x,t) + ig\phi(x,t)\chi_R(x,t)\chi_L(x,-t) \Big],
\end{aligned}
\tag{5.2}
$$

For large positive $u$ the order parameter $\phi$ has a zero vev and the time-reversal symmetry is not spontaneously broken so the low energy dynamics is controlled by the free massless Majorana fermions which is the Ising Model. By contrast, for sufficiently small $u$ the order parameter would have a nonzero vev which signals the spontaneous breaking of the time-reversal symmetry and the Majorana fermions are gapped by the Yukawa term (the last term in Equ. (5.2)) and the low energy dynamics is trivial with a nonvanishing gap. Hence for a fixed value of $g$ there is a critical value $u_c$ for $u$ where the phase transition happens and the physics is captured by a different CFT (see Fig.5 for the phase diagram). It is shown in [3] that this CFT is the Tricritical Ising Model whose central charge is $c = \frac{7}{10}$. As a result, we can use this system to engineer the RG brane that we studied in this paper. We can tune the parameter $u$ to $u_c$ first and then tune it above $u_c$ on half of the space. In practice, one could try to tune $u$ by putting the material between two plates of a capacitor in order to turn on a background electric field, which preserves the time-reversal symmetry and so should still be described by the same phase diagram.

In real materials the time-reversal symmetry breaking is achieved by magnetic ordering which sets a preferred direction of the electron spin. In the Ising Model language, this can be realized by taking into account the fermion interactions, for example dipole-dipole interactions between Cooper paired fermions. The orientation of the dipole moments can be driven by an external electric field which could potentially be used to tune to or away from the point of magnetic ordering. The advantage of this approach is that we just have to tune a single parameter to reach the TIM critical point. However, at the moment there are no clear $(1+1)$-$d$ candidate experimentally accessible systems. For example the usual systems that realize a topological superconductor of our interest are $(2+1)$-$d$ boundaries of $(3+1)$-$d$ systms, but because they sit at the extremely low temperature regime (for example superfluid $He_3$-B) it is likely to be quite challenging to use them to create $(1+1)$-$d$ boundaries of, say, $(2+1)$-$d$ films. And, despite various proposals, there are currently no known intrinsically $(2+1)$-$d$ materials that are widely accepted to be Type DIII topological superconductors. For more details of the experimental viability and difficulties we refer the readers to [3].

## 5.2 Engineering the RG Domain Wall in a Rydberg Chain

Another system that one might use to engineer the RG domain wall is the Rydberg atoms chain [24, 25]. This system consists of a one-dimensional chain of bosons (neutral atoms) with each boson as a two-level quantum mechanical system and the two energy levels are coupled to each other by a resonant laser field with a Rabi frequency $\Omega$. Nevertheless the interesting

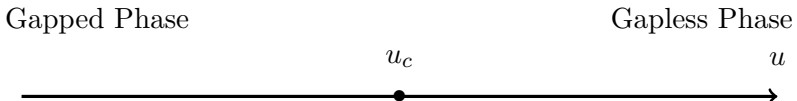

Gapped Phase                                              Gapless Phase

$u_c$                                                          $u$

**Figure 5**. *The phase diagram of the system Equ. (5.2) for a fixed value of g. At large positive u we enter a gapless phase of two Majorana fermions which is described by the Ising Model CFT. At large negative u the system is trivially gapped. Thus there is a critical value $u_c$ at which the second order phase transition happens and the physics is described by the Tricritical Ising Model CFT.*

part of system is that the Rydberg atoms are interacting if they are in the excited state and the interaction is from the dipole-dipole interaction for neaby excited atoms. Hence the whole Hamiltonian of the chain can be written as (we will follow the notations in [24])

$$H = \sum_j \left[ \frac{\Omega}{2}(b_j + b_j^\dagger) - \Delta n_j + V_1 n_j n_{j+1} + V_2 n_j n_{j+2} \right], \tag{5.3}$$

where $\Omega$ is the Rabi frequency of the external resonant laser field that couples the ground state and the excited state of each atom, $b_j^\dagger(b_j)$ is the creation (annihilation) operator for the atom at site $j$ to go from the ground (excited) state to the excited (ground) state[15], $n_j = b_j^\dagger b_j$ is the number operator at site $j$ which can be either zero or one, $\Delta$ describes the detuning from the Rydberg state, $V_1$ denotes the nearest neighbour dipole-dipole interaction and $V_2$ denotes the next-nearest neighbour dipole-dipole interaction and we have ignored higher neighbour dipole-dipole interactions which are suppressed. As [25] pointed out this system has a nice phase diagram when $V_1 = \infty$. The phase diagram is nicely plotted in [24] (see Fig.1 in [24]). Depending on the values of $\frac{V_2}{\Omega}$ and $\frac{\Delta}{\Omega}$ there are two phases of the system– a disordered phase $\langle n_j \rangle = 0$ and a $\mathbb{Z}_2$ ordered phase with $\langle n_j \rangle$ and $\langle n_{j+1} \rangle$ alternatively zero and one. There is a critical line separating the two phases which describes the phase transition between them. The interesting aspect of this critical line is that it has a tri-critical point separating the second order phase transition and the first order phase transition. This tri-critical point is described by the Tricritical Ising Model and the second order phase transition is described by the Critical Ising Model. Hence we can construct the RG domain wall in this paper by firstly tune the Rydberg chain to the tricritical point and then tune half of them away from the tricritical point to a nearby critical point for the second order phase transition between the ordered and disordered phases.

The Rydberg chain has been proposed to be a reliable platform to simulate exotic many-body spin quantum systems due to its stability and the precise tunability of the parameters [26, 27] and some of these Rydberg chains have been realized experimentally [28]. Hence the advantage of this construction is that the tuning can be done very precisely in real experimental systems of Rydberg chain and a slight disadvantage is that we have to tune multiple

---

[15]In the single particle Hilbert space at site $j$, we have $b_j = |0\rangle \langle 1|$ and $b_j^\dagger = |1\rangle \langle 0|$ so the single particle Hamiltonian $H_j = \frac{\Omega}{2}(b_j + b_j^\dagger) = \frac{\Omega}{2}(|0\rangle \langle 1| + |1\rangle \langle 0|)$ is indeed the two level coupled Hamiltonian driven by an external electric field in standard quantum mechanics.

parameters to locate the tricritical point and then tune multiple parameters simultaneously to drive half of the space away from it to a nearby critical point for the second phase transition.

## Acknowledgments

We are grateful to Rich Brower, Davide Gaiotto, Tarun Grover, Ami Katz, Severin Lüst, Rashmish Mishra, Anatoli Polkovnikov, Lisa Randall, Ruben Verresen, Ashvin Vishwanath, and Juven Wang for discussions. We would like thank Davide Gaiotto, Tarun Grover, Ami Katz and Ruben Verresen for comments on the draft. CVC and ALF are supported by the US Department of Energy Office of Science under Award Number DE-SC0015845, and the Simons Collaboration on the Non-Perturbative Bootstrap. HG is supported by the grant (272268) from the Moore Foundation "Fundamental Physics from Astronomy and Cosmology".

## A   Algebraic Construction for Minimal Models and Their Tensor Product

In this section, we review relevant results in the coset construction of unitary minimal models and their tensor product. This provides the relevant background to understand the construction of RG brane by [1] which we review in Sec.B.

### A.1   Coset Construction of Unitary Virasoro Minimal Models

Let's firstly review the coset construction of the Virasoro minimal models. Virasoro minimal models in 2d are the only unitary conformal field theories in 2d with central charge $c$ less than one. They are denoted as $\mathcal{M}_{k+3,k+2}$ with integers $k \geq 1$ and for a given $k$ the central charge is

$$c = 1 - \frac{6}{(k+2)(k+3)}\,, \tag{A.1}$$

the operator spectrum is finite and can be uniquely determined by solving the bootstrap equation.[16] Their unitarity can be understood as a consequence of existing exact unitary realization of them using the coset construction of the current algebra (i.e. the gauged Wess-Zumino-Witten (WZW) model in the Lagrangian description) as

$$\mathcal{M}_{k+3,k+2} = \frac{su(2)_k \times su(2)_1}{su(2)_{k+1}}\,. \tag{A.2}$$

Here $su(2)_k$ denotes the $su(2)$ Kac-Moody algebra at level $k$ with which we can construct a Virasoro algebra using the Sugawara construction whose central charge is

$$c_{su(2)_k} = \frac{3k}{k+2}\,. \tag{A.3}$$

---

[16]In this paper, we only consider minimal models whose operator spectrum is diagonal modular invariant. We denote such an operator content of $\mathcal{M}_{k+3,k+2}$ as $\mathcal{T}_{k+2}$.

Therefore we can check that the central charge of the coset construction from Equ. (A.2) is given by

$$c = \frac{3k}{k+2} + \frac{3}{1+2} - \frac{3(k+1)}{k+3} = 1 - \frac{6}{(k+2)(k+3)}, \tag{A.4}$$

which exactly matches that in Equ. (A.1).

Now we will spell out the map between the operator spectrums from the minimal model to that of its coset construction. A primary representation $|h,l\rangle_k$ of the current algebra $su(2)_k$ is specified by two quantum numbers $h$ and $l$ where $h$ is the conformal weight and $l$ denotes the spin-$\frac{l}{2}$ representation of the $su(2)$ algebra. The structure of the $su(2)_k$ algebra puts a constraint

$$0 \le l \le k, \tag{A.5}$$

and the conformal weight $h$ is related to $l$ as

$$h = \frac{l(l+2)}{4(k+2)}, \tag{A.6}$$

and for later convenience we will denote this representation as $(l)_k$. A primary representation $h_{r,s}(k+2)$ of the minimal model $\mathcal{M}_{k+3,k+2}$ is specified by two integer quantum numbers $r$ and $s$ which satisfies the constriants

$$1 \le r \le k+1, \quad 1 \le s \le k+2. \tag{A.7}$$

A primary representation of the tensor product current algebra $su(2)_k \times su(2)_1$ is a tensor product of the primary representations of each of them and it can be decomposed as a direct sum of the primary representations of the current algebra $su(2)_{k+1}$ and the minimal model $\mathcal{M}_{k+3,k+2}$:

$$(r-1)_k \otimes (d-1)_1 = \oplus_{\substack{0 \le (s-1) \le k+1 \\ r+s+d=1 \ \mathrm{mod}2}} \left[ (s-1)_{k+1} \otimes h_{r,s}(k+2) \right]. \tag{A.8}$$

As a result, we can denote the minimal model representation $h_{r,s}(k+2)$ using the symbols from the current algebra as $[r,d;s]$ where $1 \le r \le k+1$, $1 \le s \le k+2$, $d = 1,2$ and $r+s+d=1 \ \mathrm{mod}2$. The rule Equ. (A.8) is called the branching rule and can be used to compute the modular S-matrix of the minimal model in terms of those of the current algebra as

$$S_{[r,d;s],[r',d';s']} = S^{(k)}_{r-1,r'-1} S^{(1)}_{d-1,d'-1} S^{(k+1)}_{s-1,s'-1}, \tag{A.9}$$

where $S^{(k)}_{p,p'} = \sqrt{\frac{2}{k+2}} \sin\left(\frac{\pi(p+1)(p'+1)}{k+2}\right)$ is the modular S-matrix between the primary representations $(p)_k$ and $(p')_k$ of the $su(2)_k$ current algebra.

## A.2 Hidden Symmetry in the Tensor Product of Consecutive Minimal Models

The folding trick description of the RG brane as a conformal interface necessities the study of the direct product of two adjacent minimal models $\mathcal{M}_{k+2,k+1} \times \mathcal{M}_{k+3,k+2}$.[17] Using the coset

---

[17]A conformal interface between two CFTs can be described as a conformal boundary of the tensor product of the two CFTs if we fold the spacetime with respect to the interface.

construction Equ. (A.2) we are studying the following cosets of the current algebras

$$\mathcal{M}_{k+2,k+1} \times \mathcal{M}_{k+3,k+2} = \frac{su(2)_{k-1} \times su(2)_1}{su(2)_k} \times \frac{su(2)_k \times su(2)_1}{su(2)_{k+1}}, \tag{A.10}$$

where cancelling the common $su(2)_k$ factors in the denominator and numerator we get

$$\mathcal{M}_{k+2,k+1} \times \mathcal{M}_{k+3,k+2} = \frac{su(2)_{k-1} \times su(2)_1 \times su(2)_1}{su(2)_{k+1}}. \tag{A.11}$$

This is the relation between the apparent symmetry generators. Nevertheless, since we only consider diagonal modular invariant operator spectrum we have to be careful about whether the identity of Equ. (A.11) can be established for the operator spectrum. We denote the diagonal modular invariant spectrum of $\frac{su(2)_{k-1} \times su(2)_1 \times su(2)_1}{su(2)_{k+1}}$ as $\mathcal{T}_B$.

Now let's understand the difference between their spectrum. Using the branching rule Equ. (A.8) we can study the decomposition of the primary representations of the current algebras to those of the minimal models

$$(r-1)_{k-1} \otimes (d-1)_1 \otimes (\widetilde{d}-1)_1 = \bigoplus_{\substack{0 \leq (s-1) \leq k \\ r+s+d=1 \bmod 2}} \left[ (s-1)_k \otimes h_{r,s}(k+1) \right] \otimes (\widetilde{d}-1)_1$$

$$= \bigoplus_{\substack{0 \leq (s-1) \leq k \\ r+s+d=1 \bmod 2 \\ 0 \leq (t-1) \leq k+1 \\ s+t+\widetilde{d}=1 \bmod 2}} h_{r,s}(k+1) \otimes h_{s,t}(k+2) \otimes (t-1)_{k+1}. \tag{A.12}$$

Hence we can use the minimal model symbols to denote the primary representations of the tensor product of the current algebra $\frac{su(2)_{k-1} \times su(2)_1 \times su(2)_1}{su(2)_{k+1}}$:

$$[r,d,\widetilde{d};t] = \bigoplus_{\substack{s,r+s+d=1 \bmod 2 \\ s+t+\widetilde{d}=1 \bmod 2}} h_{r,s}(k+1) \otimes h_{s,t}(k+2), \tag{A.13}$$

where we emphasize that $d, \widetilde{d} = 1, 2$. As a result, the diagonal modular invariant spectrum $\mathcal{T}_B$ of $\frac{su(2)_{k-1} \times su(2)_1 \times su(2)_1}{su(2)_{k+1}}$ is different from the tensor product $\mathcal{T}_{k+1} \times \mathcal{T}_{k+2}$ of the diagonal modular invariant spectrums of $\mathcal{M}_{k+2,k+1}$ and $\mathcal{M}_{k+3,k+2}$. Moreover, Equ. (A.13) tells us that the (Kac-Moody) primary representations in $\mathcal{T}_B$ is only a subsector of the (Virasoro) primary representations in $\mathcal{T}_{k+1} \times \mathcal{T}_{k+2}$:

$$\mathcal{T}_B = \bigoplus_{\substack{s=1,2,\cdots,k+1 \\ r=1,2\cdots,k \\ t=1,2,\cdots,k+2}} h_{r,s}(k+1) \otimes h_{s,t}(k+2). \tag{A.14}$$

### A.2.1 The Hidden Symmetry

We have seen that the representation space $\mathcal{T}_B$ is smaller than $\mathcal{T}_{k+1} \times \mathcal{T}_{k+2}$ and this usually means that the symmetry in $\mathcal{T}_B$ is larger than those of $\mathcal{T}_{k+1} \times \mathcal{T}_{k+2}$. For a given pair of $(r,t)$ these additional symmetries will classify $h_{r,s}(k+1) \otimes h_{s,t}(k+2)$ into $[r,d,\widetilde{d};t]$ for all values

of $s$ as in Equ. (A.13). In other words, they have to classify the quantum number $s$ which takes value in $\{1, 2, \cdots, k+1\}$ into four subsets labeled by $d, \widetilde{d} = 1, 2$ as $r + s + d = 1 \mod 2$ together with $s + t + \widetilde{d} = 1 \mod 2$. This can be achieved by including the primary operators $\phi_{1,l}^{(k+1)}\phi_{l,1}^{(k+2)}$ for $\forall$ odd $l \in \{1, 2, \cdots, k+1\}$ to the current algebra.[18] Hence we see that $\mathcal{T}_B$ enjoys this enlarged chiral algebra which we denote by $\mathcal{B}$.

Interestingly, the enlarged symmetry we just uncovered for $\mathcal{T}_B$ can be further enlarged by assembling the two possible $d, \widetilde{d}$ configurations of $[r, d, \widetilde{d}; t]$ for a given pair of $r, t$[19] into a single multiplet $[r; t]$. This can be done by further including the primary operators $\phi_{1,l}^{(k+1)}\phi_{l,1}^{(k+2)}$ $\forall$ even $l \in \{1, 2, \cdots, k+1\}$ to the current algebra. Nevertheless, these new current algebra generators have half-integral conformal weights which means that they are fermionic. Using the facts that the conformal weight of the primary $\phi_{r,s}^{(k+1)}\phi_{s,t}^{(k+2)}$ is

$$\frac{(r-s)^2}{4} + \frac{(s-t)^2}{4} + \frac{r^2 - 1}{4(k+2)} - \frac{t^2 - 1}{4(k+3)}, \qquad (A.16)$$

and the fermionic currents $\phi_{1,l}^{(k+1)}\phi_{l,1}^{(k+2)}$ (i.e. $l$ is even) change the quantum number $s$ by odd integers, we can see that the change in the conformal weight Equ. (A.16) under such an action is an integer if $r + t$ is odd and a half-integer if $r + t$ is even. We can conclude that the representation $[r; t]$ is in the NS sector of this further enlarged algebra if $r + t$ is an even integer and $R$ sector if $r + t$ is an odd integer. We will denote this new chiral algebra by $\widetilde{\mathcal{B}}$.

Furthermore, we notice that there is an obvious $\mathbb{Z}_2$ symmetry in the algebra $\mathcal{B}$ which exchanges the two $su(2)_1$'s in Equ. (A.11). This symmetry will mix the representations of $\mathcal{M}_{k+2,k+1}$ and $\mathcal{M}_{k+3,k+2}$ as it is obvious from Equ. (A.10) and Equ. (A.11).

## A.2.2 Another Representation of the $\widetilde{\mathcal{B}}$ with an Explicit Supersymmetry

To make the $\mathbb{Z}_2$ symmetry manifest, we try to combine the two $su(2)_1$'s together in Equ. (A.10) by cancelling the common $su(2)_k$ factor and introducing an $su(2)_2$ factor and:

$$\mathcal{M}_{k+2,k+1} \times \mathcal{M}_{k+3,k+2} = \frac{su(2)_{k-1} \times su(2)_2}{su(2)_{k+1}} \times \frac{su(2)_1 \times su(2)_1}{su(2)_2}, \qquad (A.17)$$

which can be seen as the tensor product of an $\mathcal{N} = 1$ supersymmetric minimal model and the Ising model $\mathcal{SM}_{k+3,k+1} \times \mathcal{M}_{4,3}$. With this it is obvious that the $\mathbb{Z}_2$ symmetry exchanging the two $su(2)_1$'s is the usualy $\mathbb{Z}_2$ symmetry in the Ising model which maps the free Majorana fermion $\psi$ to $-\psi$. This can be seen by realizing that the fermion composite $\psi\overline{\psi}$ in the

---

[18]It is easy to see this using the fusion rules that $s$ is changed by an even integer under this symmetry algebra so only representations with the same $(d, \widetilde{d})$ transform to each other under this symmetry algebra. Moreover $\phi_{1,l}^{(k+1)}\phi_{l,1}^{(k+2)}$ is a legitimate current algebra as its conformal weight can be calculated

$$h = \frac{(1-l)^2}{4} + \frac{(l-1)^2}{4} + \frac{1^2 - 1}{4(k+2)} - \frac{l^2 - 1}{4(k+3)} + \frac{l^2 - 1}{4(k+3)} - \frac{1^2 - 1}{4(k+4)} = \frac{(1-l)^2}{2}, \qquad (A.15)$$

as an integer for odd $l$.

[19]Remember from Equ. (A.13) for a given $r, t$ there are only two possible configurations of $d, \widetilde{d}$.

Ising model sector (remember our operator spectrum is always diagonal modular invariant) is $\phi^{(k+1)}_{1,2}\phi^{(k+2)}_{2,1}$ which has conformal weight $h = \bar{h} = \frac{1}{2}$ from Equ. (A.16) and is invariant under the transformation $d \leftrightarrow \widetilde{d}$. This can also be seen from the fusion of two $\phi^{(k+1)}_{1,2}\phi^{(k+2)}_{2,1}$ (the result should be projected to that satisfies the constraint of operators in $\mathcal{T}_B$ i.e. of the form $\phi^{(k+1)}_{r,s}\phi^{(k+2)}_{s,t}$) which only contains two primaries $\phi^{(k+1)}_{1,1}\phi^{(k+2)}_{1,1}$ (of conformal weight $h = 0$) and $\phi^{(k+1)}_{1,3}\phi^{(k+2)}_{3,1}$ (of conformal weight $h = 2$).

The representation $[r,d,\widetilde{d};t]$ is decomposed into the representation $[r,d';t]$ in $\mathcal{SM}_{k+3,k+1}$ tensor product $[d,\widetilde{d};d']$ in the Ising model where $d' = 1,2,3$, $d + \widetilde{d} + d' = 1 \bmod 2$ and $r + d' + t = 1 \bmod 2$. This tells us that we either have the tensor product between the NS sector of $\mathcal{SM}_{k+3,k+1}$ with $1, \epsilon$ in the Ising model or the tensor product between the R sector of $\mathcal{SM}_{k+3,k+1}$ and $\sigma$ in the Ising model for representations in $\mathcal{T}_B$ (remember from the Sec. A.2.1 that $r + t$ even is the NS sector and odd is the R sector).

Moreover, it is easier to use the stress-energy tensor to figure out how the representations in one description is transformed to those in another. We are more interested in the transformation from $\mathcal{M}_{k+2,k+1} \times \mathcal{M}_{k+3,k+2}$ to $\mathcal{SM}_{k+3,k+1} \times \mathcal{M}_{4,3}$. For this purpose, we give the relations between all their chiral currents of conformal weight two:

$$T^{(k+1)} = \frac{k+3}{2k+4}T_{\mathcal{SM}} + \frac{\sqrt{(k+1)(k+3)}}{2k+4}G\psi + \frac{k-1}{2k+4}T_\psi$$

$$T^{(k+2)} = \frac{k+1}{2k+4}T_{\mathcal{SM}} - \frac{\sqrt{(k+1)(k+3)}}{2k+4}G\psi + \frac{k+5}{2k+4}T_\psi \qquad \text{(A.18)}$$

$$\phi^{(k+1)}_{1,3}\phi^{(k+2)}_{3,1} = (k+1)(k+3)T_{\mathcal{SM}} - 3\sqrt{(k+1)(k+3)}G\psi - 3(k-1)(k+5)T_\psi \,,$$

where $T_{\mathcal{SM}}$ is the holomorphic stress-energy tensor of $\mathcal{SM}_{k+3,k+1}$, $T_\psi$ is the holomorphic stress-energy tensor of the Ising model, $G$ is the superconformal current of $\mathcal{SM}_{k+3,k+1}$. From here we can see that the total stress-energy tensor $T^{(k+1)} + T^{(k+2)}$ in the $\mathcal{M}_{k+2,k+1} \times \mathcal{M}_{k+3,k+2}$ equals to the total stress-energy tensor $T_{\mathcal{SM}} + T_\psi$ in $\mathcal{SM}_{k+3,k+1} \times \mathcal{M}_{4,3}$.

For later convenience, we give the $\mathbb{Z}_2$ ($\psi \to -\psi$) transform of the two stress-energy tensors of $\mathcal{M}_{k+2,k+1} \times \mathcal{M}_{k+3,k+2}$:

$$T^{(k+1)} \to \frac{3}{(k+2)(k+4)}T^{(k+1)} + \frac{(k-1)(k+3)}{k(k+2)}T^{(k+2)} + \frac{1}{k(k+2)(k+4)}\phi^{(k+1)}_{1,3}\phi^{(k+2)}_{3,1}$$

$$T^{(k+2)} \to \frac{(k+1)(k+5)}{(k+2)(k+4)}T^{(k+1)} + \frac{3}{k(k+2)}T^{(k+2)} - \frac{1}{k(k+2)(k+4)}\phi^{(k+1)}_{1,3}\phi^{(k+2)}_{3,1} \,.$$

(A.19)

# B    Review of Gaiotto's Proposal For RG Brane

## B.1    Useful Properties of Topological Defects

Toplogical defects in a CFT are totally transmissive interfaces to the symmetry currents which can hence be arbitrarily deformed (without passing through any operators) while they

are inserted into any correlators of the CFT operators. Mathematically, we can denote a topological defect as an operator $\mathcal{D}$ which satisfies

$$[\mathcal{D}, \mathcal{J}_n] = 0, \quad [\mathcal{D}, \overline{\mathcal{J}}_n] = 0, \tag{B.1}$$

where $\mathcal{J}_n$ and $\overline{\mathcal{J}}_n$ are the holomorphic and anti-holomorphic modes of any symmetry current of the CFT. Since we only consider diagonal modular invariant CFT's, the topological defects allow Cardy's algebraic classification [29] (see [30] for a different perspective from string theory).

Hence, in a minimal model $\mathcal{M}_{k+3,k+2}$ a topological defect can be denoted as $\mathcal{D}_{r,s}^{(k+2)}$ associated to a Cardy's state $|r,s\rangle\rangle_{\text{Cardy}}$ and acts as map

$$\mathcal{D}_{r,s}^{(k+2)} = \sum_{r',s'} \frac{S_{[r;s],[r';s']}}{S_{[1;1],[r';s']}} \sum_{n} |r',s';n\rangle \langle r',s';n|, \tag{B.2}$$

in the diagonal modular invariant Hilbert space and where $S_{[r;s],[r';s']}$ is the modular S-matrix of primary representations in $\mathcal{M}_{k+3,k+2}$.[20] This description is useful for closed topological defect on a plane or a topological defect which wraps around a nontrivial cycle of a cylinder or a torus. In the former case, we get a map between operators of the (diagonal modular invariant) CFT $\mathcal{M}_{k+3,k+2}$ which maps the spinless primary operator $\mathcal{O}_{(r',s')}$ to itself with a factor $\frac{S_{[r;s],[r';s']}}{S_{[1;1],[r';s']}}$ multiplied. In the later case, it provides a specific cutting and gluing prescription along the cycle in the computation of the partition function.

The most useful property to us is that topological defects can end on certain fields called *disorder fields* [14] or *twist fields* [1]. These fields are representations of the symmetry algebra of the CFT and can in general have nonzero spin [14]. The rule is that the topological defect $\mathcal{D}_{r,s}^{(k+2)}$ can end on such an operator $\mathcal{O}_{(p,q),(m,n)}$ if $h_{r,s}(k+2)$ appears in the fusion between $h_{p,q}(k+2)$ and $h_{m,n}(k+2)$. For example, $\mathcal{D}_{r,s}^{(k+2)}$ can end on a chiral disorder operator $\phi_{r,s}^{(k+2)}$. Moreover, this tells us that we can move a topological defect $\mathcal{D}_{r,s}^{(k+2)}$ across a spinless primary field $\mathcal{O}_{(r',s')}$ and end up with a spinless disorder operator $\phi_{r',s'}^{(k+2)}$ connected to the topological defect $\mathcal{D}_{r,s}^{(k+2)}$ by a tail $\mathcal{D}_{p,q}^{(k+2)}$ such that $h_{p,q}(k+2)$ appears in the fusion between $h_{r,s}(k+2)$ and itself and also in the fusion between $h_{r',s'}(k+2)$ and itself (see. Fig.6).

As we will see this last property is useful for extracting important nonperturbative results from perturbative calculations and constraining the RG follow.

## B.2   RG Flow of the Topological Defect $\mathcal{D}_{r,1}^{(k+2)}$

The RG flow we are interested in is triggered by the spinless primary operator $\phi_{1,3}^{(k+2)}$. The transformation of the topological defect $\mathcal{D}_{r,1}^{(k+2)}$ under this RG flow can be first understood by the fact that moving $\mathcal{D}_{r,1}^{(k+2)}$ across the spinless primary operator $\phi_{1,3}^{(k+2)}$ is a trivial operation as the only representation which appears in both the fusion between $h_{r,1}(k+2)$ with itself and

---

[20]It is easy to see that Equ. (B.2) satisfies Equ. (B.1) as for example $\mathcal{J}_n = \mathcal{J}_{-n}^\dagger$.

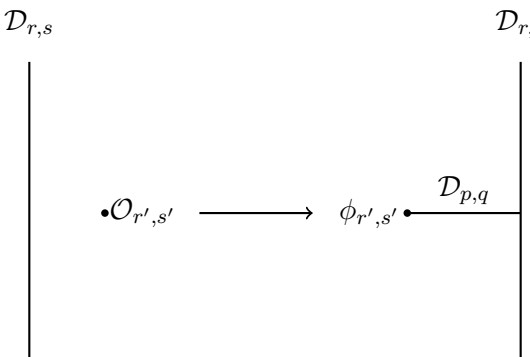

$\mathcal{D}_{r,s}$    $\mathcal{D}_{r,s}$

$\bullet\mathcal{O}_{r',s'}$ $\longrightarrow$ $\phi_{r',s'}\bullet$    $\mathcal{D}_{p,q}$

**Figure 6**. *The demonstration of crossing the topological defect $\mathcal{D}_{r,s}$ through a spinless primary operator $\mathcal{O}_{r',s'}$. This operation transforms $\mathcal{O}_{r',s'}$ to a twist field $\phi_{r',s'}$ with a topological defect $D_{p,q}$ connecting it to $D_{r,s}$.*

the fusion of $h_{1,3}(k+2)$ with itself is the trivial representation $h_{1,1}(k+2)$. Hence moving the the topological defect $\mathcal{D}_{r,1}^{(k+2)}$ across the spinless primary operator $\phi_{1,3}^{(k+2)}$ is almost a trivial operation with at most a scalar factor multiplied. Since moving $\mathcal{D}_{r,1}^{(k+2)}$ across $\phi_{1,3}^{(k+2)}$ and back is equivalent to doing nothing, the scalar factor can only be 1 or $-1$. The precise value of this scalar factor can be determined by comparing a small $D_{r,1}^{(k+2)}$ loop surrounding $\phi_{1,3}^{(k+2)}$ and a small $D_{r,1}^{(k+2)}$ loop surrounding nothing. From Equ. (B.2), we are just taking the following ratio

$$\frac{S_{[r;1],[1;3]}}{S_{[1;1],[1;3]}} \Big/ \frac{S_{[r;1],[1;1]}}{S_{[1;1],[1;1]}} = 1 \,. \tag{B.3}$$

As a result, $D_{r,1}^{(k+2)}$ is invisible to the spinless primary operator $\phi_{1,3}^{(k+2)}$ and it should be mapped to another topological defect under the RG flow. The result can be extracted from a perturbative RG flow calculation with large $k$ [16] and it is that $\mathcal{D}_{r,1}^{(k+2)}$ will flow to $\mathcal{D}_{1,r}^{(k+1)}$ if we assume that $\mathcal{M}_{k+2,k+3}$ flows to $\mathcal{M}_{k+1,k+2}$.[21]

### B.3 Extended Symmetry Algebra on RG Domain Wall from Topological Defect

When we have the RG domain wall between $\mathcal{M}_{k+3,k+2}$ on the left and $\mathcal{M}_{k+2,k+1}$ on the right, we can consider a topological defect $\mathcal{D}_{r,1}^{(k+2)}$ on the left and deforming half of it through the RG domain wall to the right (see Fig.7). The RG domain wall will transform the acrossed half to $\mathcal{D}_{1,r}^{(k+1)}$ as a result of the RG transform and we end up with a topological defect straddling between the two CFTs on the two sides of the RG brane. Now applying the folding trick, we end up with a topological defect which we denote as $\mathcal{D}_{r,1}^{(k+2)}\mathcal{D}_{1,r}^{(k+1)}$ which end on the Cardy brane (see Fig.8). We can put a chiral disorder field $\phi_{r,1}^{(k+2)}\phi_{1,r}^{(k+1)}$ at the end of $\mathcal{D}_{r,1}^{(k+2)}\mathcal{D}_{1,r}^{(k+1)}$ and push it all the way to the Cardy brane which gives us a boundary operator that doesn't change the boundary condition and have integral (for r odd) or half-integeral (for r even)

---

[21]This is also consistent with the large $k$ RG calculation [16].

conformal weight. We can do similar things for antichiral disorder fields so from the results in Sec. A.2.1 and Sec. A.2.2 we get two copies of the algebra $\widetilde{B}$ on the Cardy brane.

Gaiotto's suggestion [1] is that the Cardy brane should glue these two copies of $\widetilde{B}$ and the two copies of $\mathcal{T}_{k+1} \times \mathcal{T}_{k+2}$ inisde them are mapped to each other by the $\mathbb{Z}_2$ twisting described in Sec. A.2.2. In other words, the two copies of $\widetilde{B}$'s are glued to each other by the $\mathbb{Z}_2$ automorphism. As a result, the boundary operators should be the boundary extrapolation of $\mathbb{Z}_2$ invariant operators $\phi_{r,s}^{(k+1)}\phi_{s,t}^{(k+2)}$ with $r + t$ even.[22] This is consistent with the results from the perturbative RG calculation [15].

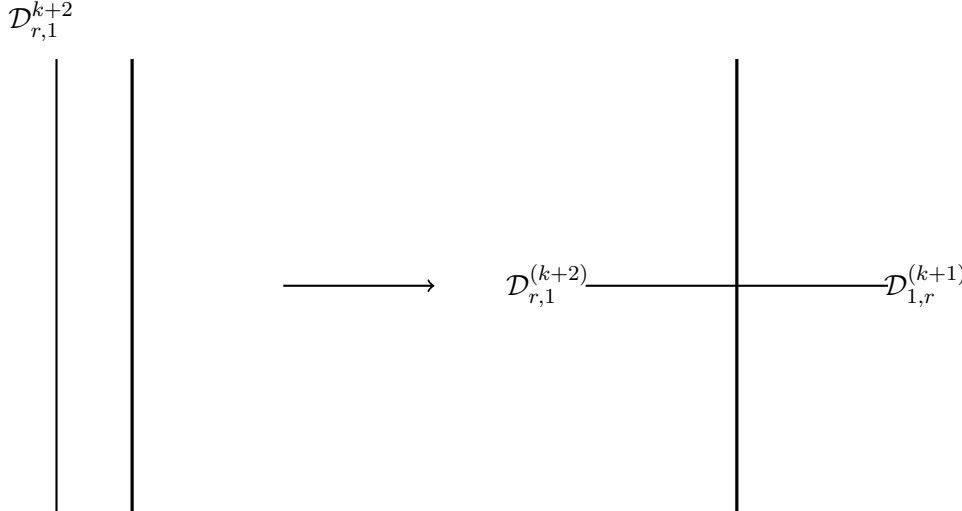

**Figure 7**. *Moving half of the topological defect $\mathcal{D}_{r,1}^{(k+2)}$ on the right through the RG brane is equivalent to performing an RG transform of of it. This gives another half of the resulting topological defect on the left as $\mathcal{D}_{1,r}^{(k+1)}$.*

## B.4 Explicit Construction of the RG Domain Wall

With the boundary operator spectrum known, a general boundary state can be written down as a linear combination of the Ishibashi states corresponding to the boundary operator spectrum under the constrains that the Verlinde formula should be satisfied by those coefficients. There are in general many such states but Gaiotto proposed the following one to be the correct one

$$\left|\widetilde{B}\right\rangle = \sum_{r,t}^{r+t\in 2\mathbb{Z}} \sqrt{S_{0,r-1}^{(k-1)} S_{0,t-1}^{(k+1)}} |r, t; \widetilde{B}\rangle\rangle, \tag{B.4}$$

where $|r, t; \widetilde{B}\rangle\rangle$ is the Ishibashi state for the algebra $\widetilde{B}$ corresponding to the representation $[r; t]$. This state is simple as it satisfies the Verlinde formula such that all the multiplicities in

---

[22]This can be seen by remember the $\mathbb{Z}_2$ exchanges the two $su(2)_1$ algebras and so it exchanges $d$ and $\widetilde{d}$ on the LHS of Equ. (A.13). As a result $\mathbb{Z}_2$ invariant operators should have $d = \widetilde{d}$ and so we get $r + t$ even from the RHS.

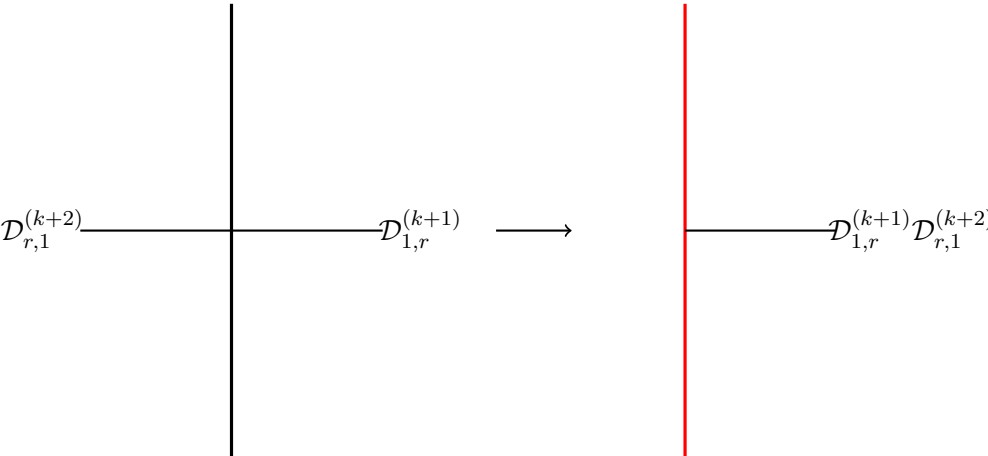

**Figure 8**. *From the unfolded picture (left) to the folded picture (right). The red wall denotes a Cardy brane in the folded picture. In the folded picture we have a topological defect $\mathcal{D}_{1,r}^{(k+1)}\mathcal{D}_{r,1}^{(k+2)}$ ending on the Cardy brane.*

the modular dual channel is uniformly one. Moreover, we can use the $\mathcal{T}_B$ language to rewrite the Ishibashi states as

$$|r,t;\widetilde{B}\rangle\rangle = \sqrt{S_{0,0}^{(1)}}|r,1,1,t;\mathcal{T}_B\rangle\rangle + \sqrt{S_{1,1}^{(1)}}|r,2,2,t;\mathcal{T}_B\rangle\rangle\,, \tag{B.5}$$

where $|r,d,\widetilde{d},t;\mathcal{T}_B\rangle\rangle$ is the Ishibashi state for the algebra $\mathcal{T}_B$ associated to the representation $[r,d,\widetilde{d};t]$ and we have used the fact that we could assemble the $\mathcal{T}_B$ representations $[r,d,\widetilde{d};t]$ for given $r$ and $t$ into the $\widetilde{B}$ representation $[r;t]$. Furthermore, using the decomposition Equ. (A.17) we can write the Ishibashi state $|r,d,\widetilde{d},t;\mathcal{T}_B\rangle\rangle$ as a superposition of tensor products of Ishibashi states in the supersymmetric minimal model and the Ising model. This is associated with the following decompositions of representations

$$[r,1,1;t] = [r,1;t] \otimes [1,1;1] + [r,3;t] \otimes [1,1;3]\,,$$
$$[r,2,2;t] = [r,1;t] \otimes [2,2;1] + [r,3;t] \otimes [2,2;3]\,. \tag{B.6}$$

However, when we translate them into the relationships between Ishibashi states we have to know the linear superposition coefficients. This can be fixed by observing that the $\mathbb{Z}_2$ automorphism we are using is localized purely in the Ising model sector. As a result, the Cardy state $\left|\widetilde{B}\right\rangle$ we get should be a tensor product of the identity Cardy state of the supersymmetric minimal model $\mathcal{SM}_{k+3,k+1}$ and a nontrivial Cardy state of the Ising model whose Ishibashi components are $|1,1;1\rangle\rangle$, $|1,1;3\rangle\rangle$, $|2,2;1\rangle\rangle$ and $|2,2;3\rangle\rangle$ (i.e. $|\epsilon\rangle\rangle$ and $|1\rangle\rangle$ by the branching rule Equ. (A.8)). This Cardy state is $|\sigma\rangle = \frac{1}{\sqrt{2}}(|1\rangle\rangle - |\epsilon\rangle\rangle)$ which indeed implements the $\mathbb{Z}_2$

transformation $\psi \to -\psi$.[23] Hence, the Cardy state Equ. (B.4) can be written as

$$\left|\widetilde{B}\right\rangle = |1\rangle_{\text{NS}} \otimes |\sigma\rangle_{\text{Ising}} = \frac{1}{\sqrt{2}} |1\rangle_{\text{NS}} \otimes (|1\rangle\rangle - |\epsilon\rangle\rangle),\tag{B.7}$$

where the Cardy state $|1\rangle_{\text{NS}}$ can be written in the Ishibashi states as

$$|1\rangle_{\text{NS}} = \sum_{r,t}^{r+t\in 2\mathbb{Z}} \sqrt{S_{[r;t],[0,0]}}|r;t\rangle\rangle$$

$$= \sum_{r,t}^{r+t\in 2\mathbb{Z}} \sqrt{S_{0,r-1}^{(k-1)}S_{0,t-1}^{(k+2)}S_{0,0}^{(2)}}|r,1;t\rangle\rangle + \sum_{r,t}^{r+t\in 2\mathbb{Z}} \sqrt{S_{0,r-1}^{(k-1)}S_{0,t-1}^{(k+2)}S_{0,2}^{(2)}}|r,3;t\rangle\rangle \tag{B.8}$$

$$= \frac{1}{\sqrt{2}} \sum_{r,t}^{r+t\in 2\mathbb{Z}} \sqrt{S_{0,r-1}^{(k-1)}S_{0,t-1}^{(k+2)}}\left(|r,1;t\rangle\rangle + |r,3;t\rangle\rangle\right).$$

Here we notice that $|r,1;t\rangle\rangle$ and $|r,3;t\rangle\rangle$ (for $r+t\in 2\mathbb{Z}$) form an NS sector supermultiplet.

## B.5 Computation of One-Point Functions

So far we have constructed the RG brane between two consecutive minimal models $\mathcal{M}_{k+2,k+1}$ and $\mathcal{M}_{k+3,k+2}$ as a rational brane by embedding the diagonal modular invariant representations $h_{r,s}(k+1)\otimes h_{s,t}(k+2) = [r,d;s]\otimes[s,\widetilde{d};t]$ of the algebra $\mathcal{M}_{k+2,k+1}\times\mathcal{M}_{k+3,k+2}$ into the diagonal modular invariant representations $[r,d,\widetilde{d};t]$ of an equivalent algebra $\frac{su(2)_{k-1}\times su(2)_1\times su(2)_1}{su(2)_{k+1}}$. For simplicity we will called the former theory $\mathcal{T}_{\mathcal{A}}$ (i.e. $\mathcal{T}_{k+1}\times\mathcal{T}_{k+2}$) and the latter theory $\mathcal{T}_{\mathcal{B}}$. The precise value of the one-point functions of the operators $\phi_{r,s}^{(k+1)}\phi_{s,t}^{(k+2)}$ in $\mathcal{T}_{\mathcal{A}}$ is of important physical relevance as they tell us how the operators $\phi_{r,s}^{(k+2)}$ and $\phi_{s,t}^{(k+1)}$ are mixed under the RG flow. To find the precise value of these one-point functions we have to map the rational brane Equ. (B.7) we have constructed in $\mathcal{T}_{\mathcal{B}}$ back to $\mathcal{T}_{\mathcal{A}}$ and then the one-point functions can be easily computed.

However, the map of the rational brane Equ. (B.7) to $\mathcal{T}_{\mathcal{A}}$ is not trivial. This can be achieved by firstly constructing a proper $\mathcal{T}_{\mathcal{A}}$ topological interface (i.e. totally transmissive for symmetry currents in $\mathcal{T}_{\mathcal{A}}$) $\mathcal{I}_1$ separating $\mathcal{T}_{\mathcal{A}}$ and $\mathcal{T}_{\mathcal{B}}$ and then fusing it with the $\mathcal{T}_{\mathcal{B}}$ rational brane Equ. (B.7) (see Fig.9). A proper $\mathcal{I}_1$ should allow the $\mathcal{T}_{\mathcal{A}}$ topological defects

$$\mathcal{D}^{\mathcal{A}}_{[r,d;s]\otimes[s,\widetilde{d};t]} = \sum_{r',d',s',\widetilde{d}',t'} \frac{S_{[r,d;s]\otimes[s,\widetilde{d};t],[r',d';s']\otimes[s',\widetilde{d}',t']}}{S_{[1,1;1]\otimes[1,1;1],[r',d';s']\otimes[s',\widetilde{d}',t']}} \sum_n \left|r',d',s',\widetilde{d}',t';n\right\rangle\left\langle r',d',s',\widetilde{d}',t';n\right|,\tag{B.9}$$

---

[23]This can be seen by using the doubling trick, identifying the Cardy boundary $|\sigma\rangle$ as a topological defect $\mathcal{D}_\sigma$ in the resulting chiral theory and using the fact that $\mathcal{D}_\sigma$ is a group like defect.

to end on it.[24] This would be ensured if $D^{\mathcal{A}}_{[r,d;s]\otimes[s,\widetilde{d};t]}$ appears in the fusion of $\overline{\mathcal{I}_1}$ and $\mathcal{I}_1$ or in other words if $\mathcal{I}_1\overline{\mathcal{I}_1}$ is a direct sum of the topological defects $D^{\mathcal{A}}_{[r,d;s]\otimes[s,\widetilde{d};t]}$ (see Fig.10). This is satisfied by the following construction of $\mathcal{I}_1$

$$
\mathcal{I}_1 = \sum_{r,t,s,d,\widetilde{d}} \sqrt{\frac{S_{[1;1],[r;t]}}{S_{[1,1;1]\otimes[1,1;1],[r,d;s]\otimes[s,\widetilde{d};t]}}} \sum_n \left| r,d,s,\widetilde{d},t;n \right\rangle \left\langle r,d,s,\widetilde{d},t;n \right| , \tag{B.11}
$$

where the operator $\left\langle r,d,s,\widetilde{d},t;n \right|$ are orthonormal $\mathcal{T}_{\mathcal{A}}$ descendents of the primary states $\left\langle r,d,s,\widetilde{d},t \right|$. The operator $\sum_n \left| r,d,s,\widetilde{d},t;n \right\rangle \left\langle r,d,s,\widetilde{d},t;n \right|$ acts nontrivially only on the Ishibashi state $|r,t;\widetilde{B}\rangle\rangle$ and maps it to the Ishibashi state $|r,d,s,\widetilde{d},t;\mathcal{A}\rangle\rangle$.[25]

Moreover, it is easy to check that we have

$$
\begin{aligned}
\mathcal{I}_1\overline{\mathcal{I}_1} &= \sum_{r,t,s,d,\widetilde{d}} \frac{S_{[1;1],[r;t]}}{S_{[1,1;1]\otimes[1,1;1],[r,d;s]\otimes[s,\widetilde{d};t]}} \sum_n \left| r,d,s,\widetilde{d},t;n \right\rangle \left\langle r,d,s,\widetilde{d},t;n \right| , \\
&= \sum_{s',d',\widetilde{d}'} \mathcal{D}^{\mathcal{A}}_{[1,d';s']\otimes[s',\widetilde{d}';1]} ,
\end{aligned} \tag{B.12}
$$

where the last step comes from the fact that

$$
\sum_{s',d',\widetilde{d}'} S_{[1,d';s']\otimes[s',\widetilde{d}';1],[r,d;s]\otimes[s,\widetilde{d};t]} = S_{[1;1],[r;t]} . \tag{B.13}
$$

This relation can be obtained from the following considerations. Let's consider the modular character $\chi_{[r;t]}(\tau)$ of the $\widetilde{B}$ representation $[r;t]$. Since we know that $[r;t]$ can be split into $\mathcal{A}$ representations $[r,d;s]\otimes[s,\widetilde{d};t]$ with multiplicities as one (see Sec.A.2.1) so we have

$$
\chi_{[r;t]}(\tau) = \sum_{s,d,\widetilde{d}} \chi_{[r,d;s]\otimes[s,\widetilde{d};t]}(\tau) . \tag{B.14}
$$

Now we consider $r=1, t=1$, do a modular transform

$$
S: \quad \tau- \to -\frac{1}{\tau} , \tag{B.15}
$$

---

[24]We of course have the following constraints

$$
\begin{aligned}
r+d+s &=1 \text{ mod2}, \quad s+\widetilde{d}+t = 1 \text{ mod2}, \\
r'+d'+s' &=1 \text{ mod2}, \quad s'+\widetilde{d}'+t' = 1 \text{ mod2}.
\end{aligned} \tag{B.10}
$$

[25]This is because $[r;t]$ can be split into $\mathcal{A}$ representations $[r,d;s]\otimes[s,\widetilde{d};t]$ with multiplicities as one (see Sec.A.2.1).

and we will have

$$\chi_{[1;1]}(\tau) = \sum_{r,t} S_{[1;1],[r;t]}\chi_{[r;t]}(-\frac{1}{\tau}) = \sum_{r,t,s,d,\widetilde{d}} S_{[1;1],[r;t]}\chi_{[r,d;s]\otimes[s,\widetilde{d};t]}(-\frac{1}{\tau})$$

$$= \sum_{s',d',\widetilde{d}'} \chi_{[1,d';s']\otimes[s',\widetilde{d}';1]}(\tau) = \sum_{s',d',\widetilde{d}'} \sum_{r,t,s,d,\widetilde{d}} S_{[1,d';s']\otimes[s',\widetilde{d}';1],[r,d;s]\otimes[s,\widetilde{d};t]}\chi_{[r,d;s]\otimes[s,\widetilde{d};t]}(-\frac{1}{\tau}),$$

(B.16)

which gives us Equ. (B.13) if we compare the end of the first line and the end of the second line. Furthermore, Equ. (B.12) tells us that the $\mathcal{T}_{\mathcal{A}}$ defects $\mathcal{D}^{\mathcal{A}}_{[1,d,r]\otimes[r,\widetilde{d},1]} = \mathcal{D}^{(k+2)}_{r,1}\mathcal{D}^{(k+1)}_{1,r}$ can end on the defect $\mathcal{I}_1$ and hence could end on the Cardy boundary $\mathcal{I}_1|\widetilde{B}\rangle\rangle$ after we fuse $\mathcal{I}_1$ with $|\widetilde{B}\rangle\rangle$ getting a Cardy state for $\mathcal{T}_{\mathcal{A}}$. This is precisely what we used in the construction of the RG domain wall as we discussed in Sec.B.3 (see Fig.11).

Now we can use the map Equ. (B.11) to map the boundary state Equ. (B.4) to a boundary state in the $\mathcal{T}_{\mathcal{A}}$ theory (see Fig.9)

$$|\mathcal{A}\rangle\rangle = \mathcal{I}_1|\widetilde{B}\rangle\rangle = \sum_{r,t,s,d,\widetilde{d},d'}^{r+t\in 2\mathbb{Z}} \alpha_{r,d,s,\widetilde{d},t,d'}\sqrt{S^{(k-1)}_{0,r-1}S^{(k+1)}_{0,t-1}S^{(1)}_{0,d-1}S^{(1)}_{0,\widetilde{d}-1}}\sqrt{\frac{S_{[1;1],[r;t]}}{S_{[1,1;1]\otimes[1,1;1],[r,d;s]\otimes[s,\widetilde{d},t]}}}|r,d,s,\widetilde{d},t\rangle\rangle,$$

(B.17)

where $|r,d,s,\widetilde{d},t\rangle\rangle$ is the $\mathcal{T}_{\mathcal{A}}$ Ishibashi state associated with the representation $[r,d;s]\otimes[s,\widetilde{d};t]$ and here we emphasize that the constrains $r + d + s = 1 \mod 2$, $s + \widetilde{d} + t = 1 \mod 2$ and $d + \widetilde{d} + d' = 1 \mod 2$ should be satisfied. The coefficient $\alpha_{r,d,s,\widetilde{d},t,d'}$ can be figured out by normalization of the state and the decomposition Equ. (A.18). For the $(r,s,t,d,\widetilde{d})$ such that $d'$ can be uniquely determined this factor is just one. This tells us that in these cases the one-point function of the operator $\phi^{(k+1)}_{r,s}\phi^{(k+2)}_{s,t}$ is given by

$$\sqrt{S^{(k-1)}_{0,r-1}S^{(k+1)}_{0,t-1}S^{(1)}_{0,d-1}S^{(1)}_{0,\widetilde{d}-1}}\sqrt{\frac{S_{[1;1],[r;t]}}{S_{[1,1;1]\otimes[1,1;1],[r,d;s]\otimes[s,\widetilde{d},t]}}} = \frac{\sqrt{S^{(k-1)}_{0,r-1}S^{(k+1)}_{0,t-1}}}{S^{(k)}_{0,s-1}}\delta_{d,\widetilde{d}}.$$

(B.18)

Examples of more general cases that $d'$ is not uniquely fixed can be found in Equ. (3.21) and Equ. (4.10).

## C  Topological Superconductors and Majorana Fermions

In this appendix we will illustrate the gist of topological superconductors for readers with high-energy physics background. We will consider a toy Hamiltonian for the Type IIID topological superconductor which we will use to illustrate the relevant physical background of Sec. 5.1.

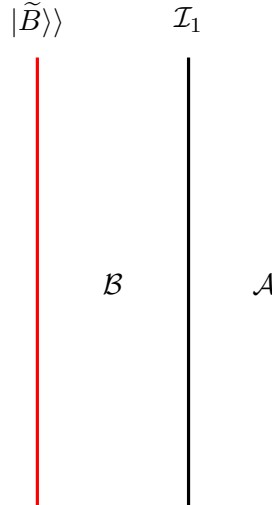

**Figure 9**. *The picture in the folded description of mapping the $\mathcal{T}_\mathcal{B}$ Cardy state to a $\mathcal{A} \subset \mathcal{B}$ theory Cardy state by fusing it with a $\mathcal{T}_\mathcal{A}$ topological defect $\mathcal{I}_1$ that separate the $\mathcal{A}$ theory and the $\mathcal{B}$ theory. The fusion is topologically achieved by pushing $\mathcal{I}_1$ all the way to $|\widetilde{B}\rangle\rangle$.*

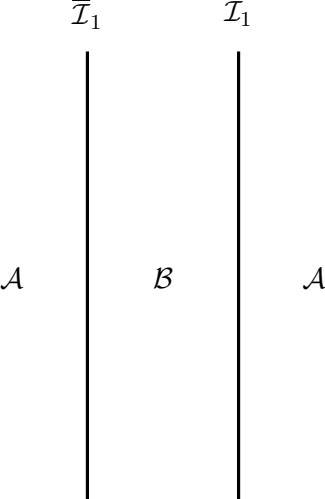

**Figure 10**. *The picture in of fusing $\mathcal{I}_1$ with $\overline{\widetilde{\mathcal{I}}}_1$. The fusion is achieved topologically by pushing $\mathcal{I}_1$ all the way to be coincident with $\overline{\widetilde{\mathcal{I}}}_1$. The result is a $\mathcal{T}_\mathcal{A}$ topological defect.*

## C.1  A Toy Hamiltonian and Its Topological Properties

Let's consider a two-dimensional electronic system which hosts electrons and holes. A typical such system is described by a lattice Hamiltonian where we have a fermionic degree of freedom on each lattice site. The feromions can be of spin-up and spin-down. We consider such a

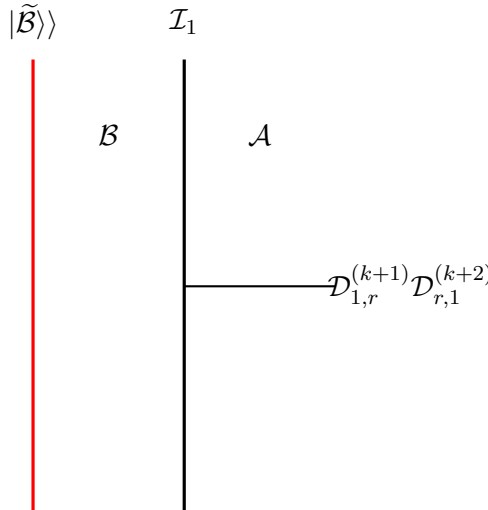

**Figure 11**. *The demonstration in the folded picture that how we could have $\mathcal{T}_A$ topological defects $\mathcal{D}_{1,r}^{(k+1)}\mathcal{D}_{r,1}^{(k+2)}$ ending on the RG brane which is the fusion of $\mathcal{I}_1$ with $|\widetilde{\mathcal{B}}\rangle\rangle$. The fusion is achieved topological by pushing $\mathcal{I}_1$ all the way to the $\mathcal{T}_{\mathcal{B}}$ Cardy boundary $|\widetilde{B}\rangle\rangle$.*

system in momentum space with a p-wave superconducting pairing

$$H = \frac{1}{2} \sum_{\vec{p} \in BZ} H_{\vec{p}}$$

$$= \frac{1}{2} \sum_{\vec{p} \in BZ} \left[ \psi_{\vec{p},\uparrow}^{\dagger}(\frac{p^2}{2m} - \mu)\psi_{\vec{p},\uparrow} + \Delta\psi_{-\vec{p},\uparrow}(p_x + ip_y)\psi_{\vec{p},\uparrow} + \psi_{\vec{p},\downarrow}^{\dagger}(\frac{p^2}{2m} - \mu)\psi_{\vec{p},\downarrow} + \Delta\psi_{-\vec{p},\downarrow}(p_x - ip_y)\psi_{\vec{p},\downarrow} + H.C. \right],$$

$$(C.1)$$

where $\psi_{\vec{p},s}^{\dagger}$ is a creation operator for electron with momentum $\vec{p}$ and spin $s$ ($s = \uparrow, \downarrow$), $\psi_{\vec{p},s}$ is the corresponding creation operator for a hole, $\mu$ is the chemical potential, $\Delta$ is the p-wave pairing parameter and the momentum $\vec{p}$ lives in the Brillouin zone. This is called the Bogolubov-de Genes Hamiltonian in condensed matter literature and the way the property of an electronic material is analyzed is to firstly write down the Bogolubov-de Genes Hamiltonian for that material and then solve for the eigenenergy and eigenmodes. These eigenmodes are called quasi-particles. We don't intend to solve for the eigenmodes and eigenenergy of the Hamiltonian Equ. (C.1) but we will analyze some important properties of it that tells us nontrivial information of the spectrum and phases for the electronic system that is described by this Hamiltonian.

Firstly, this Hamiltonian transforms under the particle-hole symmetry

$$\mathcal{P} : \psi_{\vec{p},s} \to \psi_{-\vec{p},-s}^{\dagger}, \quad \psi_{\vec{p},s}^{\dagger} \to \psi_{-\vec{p},-s} \tag{C.2}$$

as

$$PH_{\vec{p}}P = -H_{-\vec{p}}. \quad PHP = -H. \tag{C.3}$$

Hence, the spectrum of this Hamiltonian is paired into pairs of opposite values

$$H = \sum_{E \geq 0} \left[ E \gamma_E^\dagger \gamma_E - E \gamma_{-E}^\dagger \gamma_{-E} \right] , \tag{C.4}$$

where $\gamma_E^\dagger$ and $\gamma_E$ are the creation and annihilation operators for the quasi-particles. Moreover, the spin-up and spin down degress of freedom don't mixed in the Hamiltonian. We can analyze spectrum of $H_{\vec{p}}$. This can be down by defining

$$\Psi_{\vec{p},\uparrow}^\dagger = (\psi_{\vec{p},\uparrow}^\dagger, \psi_{-\vec{p},\uparrow}) , \quad \Psi_{\vec{p},\downarrow}^\dagger = (\psi_{\vec{p},\downarrow}^\dagger, \psi_{-\vec{p},\downarrow}) . \tag{C.5}$$

We can see that in both the spin-up and spin-down sector we have eigenvalues $E_\pm(\vec{p}) = \pm\sqrt{(\frac{p^2}{2m} - \mu)^2 + \Delta^2 p^2}$. Hence, there is a degeneracy of each eigenvalue $E$ and this degeneracy is due to the spin quantum number. As a result, we can see that when $\mu = 0$ there is a gapless zero mode sector for $\vec{p} = 0$ and close to this zero mode sector there is an emergent time-reversal symmetry:

$$\mathcal{T} : \psi_{\vec{p},s} \to \psi_{-\vec{p},-s} , \quad \psi_{\vec{p},s}^\dagger \to \psi_{-\vec{p},-s}^\dagger , \quad \vec{p} \to -\vec{p} , \quad i \to -i , \tag{C.6}$$

which can be seen from Equ. (C.1) by taking $\mu = 0$ and ignoring $p^2$ terms and this symmetry protects the gap.

Secondly, when $\mu \neq 0$, there is no zero modes and there is always a gap in the spectrum. We want to understand the difference between $\mu > 0$ and $\mu < 0$ cases. Since the two spin sectors don't couple, we can focus on the spin-up sector for which we can write the Hamiltonian as (for simplicity we will take $\Delta = 1$ hereafter)

$$H_{\vec{p},\uparrow} + H_{-\vec{p},\uparrow} = \Psi_{\vec{p},\uparrow}^\dagger \begin{pmatrix} \frac{p^2}{2m} - \mu & p_x + ip_y \\ p_x - ip_y & -(\frac{p^2}{2m} - \mu) \end{pmatrix} \Psi_{\vec{p},\uparrow} = \Psi_{\vec{p},\uparrow}^\dagger \vec{h}(\vec{p}) \cdot \vec{\sigma} \Psi_{\vec{p},\uparrow} , \tag{C.7}$$

where we have $\vec{h}(\vec{p}) = (p_x, -p_y, \frac{p^2}{2m} - \mu)$. As long as $\mu \neq 0$, the vector $\vec{h}(\vec{p})$ is never zero, so we can normalize it and define

$$\widehat{h}(\vec{p}) = \frac{\vec{h}(\vec{p})}{|\vec{h}(\vec{p})|} , \tag{C.8}$$

and consider the Chern-number of the map $\widehat{h}(\vec{p})$ (notice that $\widehat{h}(\vec{p}) \to (0,0,1)$ as $\vec{p} \to \infty$)

$$C = \int \frac{d^2\vec{p}}{4\pi} \left[ \widehat{h}(\vec{p}) \cdot \left( \partial_{p_x} \widehat{h}(\vec{p}) \times \partial_{p_y} \widehat{h}(\vec{p}) \right) \right] , \tag{C.9}$$

which determines the winding number of the map $\widehat{h}(\vec{p})$. We can see that $h^z(\vec{p})$ behaves rather differently when $\mu > 0$ and $\mu < 0$. As a result, in the former case as $\vec{p}$ goes from 0 to $\infty$ $\widehat{h}(\vec{p})$ starts with pointing to the South pole and ends up with pointing to the North pole and in the later case $\widehat{h}^z(\vec{p})$ never takes a negative value so it always points to a point on the Northern hemisphere. That is that in the $\mu > 0$ case the winding number $C = 1$ and in the $\mu < 0$ case $C = 0$.

In summary, the two gapped phases $\mu > 0$ and $\mu < 0$ are different topologically and their boundary $\mu = 0$ has gapless zero modes degenerate in spin and enjoy an emergent time-reversal symmetry in the low energy regime. Moreover, we can see that the $\mu < 0$ phase is a trivially gapped phase (this can be seen by taking $\mu \to -\infty$) and $\mu > 0$ is a weakly gapped phase which is topological as it has a nontrivial topological number $C = 1$.

## C.2   Localized Majorana Modes on the Boundary– The Ising Model

Now we want to consider, the system described by the Hamiltonian Equ. (C.1) on a manifold with boundary. For example a finite piece of the electronic material whose physics is described by Equ. (C.1). We are interested in the physics of its boundary when we tune its bulk to the topologically nontrivial phase $\mu > 0$. This can be done in experiment by electron doping.

At a fine-grained level we have to specify the precise boundary conditions and solve for the energy spectrum of the Hamiltonian under this boundary conditions. Nevertheless, we can grasp the gist using a slightly coarse-grained model for the boundary. We can think of the environment as in the trivial phase $\mu < 0$ of the Hamiltonian Equ. (C.1). Since the bulk of the system has been tuned to the topologically nontrivial phase $\mu > 0$, the boundary can be thought of as a thin buffer zone between these two phases where $\mu = 0$. For simplicity we will take the $y-$direction to be the direction normal to the boundary of the material i.e. the buffer zone is thin in the $y-$direction. We are interested in the low energy physics in the buffer zone. In the low energy regime where the spin-up sector Hamiltonian can be written as

$$H_{\vec{p},\uparrow} + H_{-\vec{p},\uparrow} = \Psi^\dagger_{\vec{p},\uparrow} \begin{pmatrix} 0 & p_x + ip_y \\ p_x - ip_y & 0 \end{pmatrix} \Psi_{\vec{p},\uparrow}. \tag{C.10}$$

The buffer zoom is extremely thin in the $y-$direction so we can set $p_y$ to zero in the low energy regime. Expanding the matrix representation, this gives us the Hamiltonian

$$H_\uparrow = \sum_{p_x \geq 0} p_x (\psi^\dagger_{p_x,\uparrow} \psi^\dagger_{-p_x,\uparrow} + \psi_{-p_x,\uparrow} \psi_{p_x,\uparrow}). \tag{C.11}$$

Let's do the following redefinition

$$\psi_{p_x,\uparrow} = \frac{1}{\sqrt{2}} (\chi^\dagger_{p_x,\uparrow} + \chi_{-p_x,\uparrow}), \tag{C.12}$$

where $\chi_{p_x,\uparrow}$ is a fermionic annihilation operator satisfies the standard algebra with the creation operator. This gives us

$$H_\uparrow = \sum_{p_x \geq 0} p_x (\chi_{p_x,\uparrow} \chi_{-p_x,\uparrow} + \chi_{p_x,\uparrow} \chi^\dagger_{p_x,\uparrow} + \chi^\dagger_{-p_x,\uparrow} \chi_{-p_x,\uparrow} + \chi^\dagger_{-p_x,\uparrow} \chi^\dagger_{p_x,\uparrow})$$

$$= \frac{1}{2} \sum_{p_x} p_x (-\chi_{-p_x,\uparrow} \chi_{p_x,\uparrow} + \chi_{p_x,\uparrow} \chi^\dagger_{p_x,\uparrow} - \chi^\dagger_{p_x,\uparrow} \chi_{p_x,\uparrow} + \chi^\dagger_{-p_x,\uparrow} \chi^\dagger_{p_x,\uparrow}) + const. \tag{C.13}$$

$$= \frac{1}{2} \int dx \chi_\uparrow(x,t) i\partial_x \chi_\uparrow(x,t),$$

where we have a Majorana field

$$\chi_\uparrow(x,t) = \sum_{p_x} (\chi_{p_x,\uparrow} e^{ip_x(x+t)} + \chi_{p_x,\uparrow}^\dagger e^{-ip_x(x+t)}) \,. \tag{C.14}$$

The Majorana field satisfies $\chi_\uparrow(x,t) = \chi_\uparrow^\dagger(x,t)$. So we have the Lagrangian

$$L_\uparrow = \frac{i}{2} \int dx \chi_\uparrow(x,t)(\partial_t - \partial_x)\chi_\uparrow(x,t) \,, \tag{C.15}$$

which is chiral. Similarly, in the spin-down sector we have the Lagrangian

$$L_\downarrow = \frac{i}{2} \int dx \chi_\downarrow(x,t)(\partial_t + \partial_x)\chi_\downarrow(x,t) \,, \tag{C.16}$$

where

$$\psi_{p_x,\downarrow} = \frac{i}{\sqrt{2}} (\chi_{p_x,\downarrow}^\dagger + \chi_{-p_x,\downarrow}) \,. \tag{C.17}$$

In total we have the Lagrangian

$$L = \frac{i}{2} \int dx \Big[\chi_\downarrow(x,t)(\partial_t + \partial_x)\chi_\downarrow(x,t) + \chi_\uparrow(x,t)(\partial_t - \partial_x)\chi_\uparrow(x,t)\Big] \,, \tag{C.18}$$

which describes two chiral Majorona fermions and the time-reversal symmetry Equ. (C.19) translates to

$$\mathcal{T} : \chi_\uparrow(x,t) \to -i\chi_\downarrow(x,-t) \,, \quad \chi_\downarrow(x,t) \to i\chi_\uparrow(x,-t) \,, \quad i \to -i \,. \tag{C.19}$$

Hence we have

$$\mathcal{T}^2 = -1 \,. \tag{C.20}$$

This time reversal symmetry prevents the Majorana ferimons to have Majorana mass term $im\chi_\downarrow(x,t)\chi_\uparrow(x,t)$. As a result, we have a gapless Majorana fermion whose mass is protected by the time-reversal symmetry Equ. (C.19) and this theory is localized in the thin buffer zone or equivalent on the boundary of the bulk topological material. This theory is nothing but the 1+1-dimensional Ising Model. We emphasized that Equ. (C.20) is an important character of the Type IIID topological superconductor which ensure the boundary of the nontrivial topological phase to host two anti-propagating Majorana fermions and hence gives us the Ising Model.

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
