# Peer review of "CFT and Lattice Correlators Near an RG Domain Wall between Minimal Models"

_SciPost Physics_

## Round 1 · Referee Report · Anonymous · 2023-11-18

Report
The authors apply general results of Gaiotto to study the specific example of an RG domain wall between the Ising and tricritical Ising CFTs. They find explicit formulas for several correlators, and then compare them to DMRG numerics on a lattice spin-chain realization. The agreement is excellent.
This paper contains no major advances in the state of the art, as the construction is Gaiotto's, and most of the technicalities needed for this example were worked out long ago by Crnkovic et al. Nonetheless, I think the RG domain wall is interesting enough to make it worth doing an explicit example like this. Formal constructions are often not checked numerically like this paper does.
There is one possible flaw in the analysis, however. I don't understand their eqn 3.6. Namely, they factorize the Ising spin field in to holo x antiholomorphic parts, which in general is very wrong: it is the *sum* of such factorizations. Possibly in this explicit example such a factorization holds, but the authors need to explain why and to point out that in general this is not possible.
For the most part, the paper is very clear, with a nice review of Gaiotto's construction in the appendices. However, one key technical point is not addressed well, see requested change 2 below. And while I wholeheartedly encourage people coming from a more formal background to pay attention to how experiment may be relevant, section 5 doesn't say anything that the earlier works don't. For example, they don't give any serious insight into how the experimentalists would implement a spatially inhomogeneous coupling (I don't understand their capacitor comment). In general, even a one parameter tuning to the TCI point will be rather difficult in practice, much less over only part of the system.
Provided the authors fix the issue with eqn 3.6, I support publication in SciPost.
Requested changes
1 - Absolutely must justify explicitly the factorization in eqn (3.6)
2 - Need to explain much more clearly the decomposition of the correlator (3.9). I realize this is a result of Crnkovic et al, but this is really central to their calculation. Thus they need to write it so that someone not familiar with the old paper can follow it.
3 - for the "experimental" section 5, they should make it clearer that they are just reviewing earlier proposals (or if there is something novel I missed, they should indicate it).
4 - I presume the lines in figure 4 are the theory curves? If so, they should say it.
5 - The authors should take a look at https://arxiv.org/abs/0911.4969 , as this paper analyzes the same flow the authors do. Possibly something interesting could be learned from the integrability.
Anonymous on 2023-10-24 [id 4057]
Please provide a DOI for all references whenever possible, as requested by the SciPost manuscript preparation guidelines. In addition, e.g., the information in Ref. [25] is incomplete. You might also consider using the SciPost template for submissions to SciPost Physics, see https://scipost.org/SciPostPhys/authoring#manuprep for further details.

---

## Round 1 · Referee Report · Anonymous · 2023-12-10

Strengths
- a detailed CFT calculation for the RG domain wall between the Ising and the tricritical Ising CFT.
- a numerical test of these predictions for a specific microscopic model exhibiting Ising and tricritical Ising CFT low-energy descriptions.
Weaknesses
- the numerical setup lacks a detailed description of the geometry, such that the precise location of the operators and the spatial symmetries for certain correlation positions are unclear.
- some numerical calculations show small, but clearly visible discrepancies between the CFT prediction and the numerical results, without being noticed and discussed in the text.
Report
This is an interesting paper which works out detailed predictions for the behaviour of correlation functions at an RG domain wall between the critical and the tricritical Ising model, and then test these predictions with a DMRG simulations of a microscopic model which can be tuned to either of the two targeted CFTs.
I focus here on the numerical part. I appreciate the overall agreement between the CFT predictions and the numerics, but I would like to see a more detailed description of the lattice setup and its relation to CFT coordinates. The reason is that the mirror symmetry of the blue CFT curves is not respected by the DMRG data, and I (and perhaps other readers) would like to understand the reason for this asymmetry. Furthermore the authors should also check the legend of the figure, I am not sure the angles theta_1 are correctly labeled.
It seems to me that the agreement between the CFT predictions and the DMRG data is less convincing for the pure Ising CFT (Fig 3 right) and the Ising CFT part of the RG domain wall. The authors should discuss possible reasons for these discrepancies.
Requested changes
see main report.

---

## Round 1 · Referee Report · Anonymous · 2023-12-12

Strengths
1- Interesting test of CFT predictions
2- Detailed CFT derivations
Weaknesses
1- numerical analysis very basics
2- lack of finite size scaling
Report
I have read the paper and found it interesting. It is aimed at studying the RG domain wall between to CFTs. They consider the simplest scenario of an interface between the Ising model and the Tricrtical Ising model.
They compute the correlations functions between primary operator in the CFT and compare the results they find with DMRG simulations.
They also discuss about possible experimental implementations.
As mentioned I find the paper interesting, but I am struggling to judge the relevance of it
I cannot really judge the CFT part of the calculations, and their difficulty/novelty, but both the numerical results and the experimental proposals are not very detailed.
The numerical analysis seems to confirm the analytical predictions but with clear deviations that are not discussed. Not only in the UV (as mentioned in the discussion) but also in the infra-red (for very large separations for |i-j| between 50 and 100 in Fig 3 IM).
In such a case I would have expected some attempt of finite size scaling to see if things converge to what expected once higher order corrections are taken into account, but there is nothing of this kind in the paper.
Also there is no mention on how PBC have been implemented. In particular DMRG is known to work purely with PBC and one typically needs to use a gradient descent rather than simple DMRG in order to obtain good results.
The authors have used an open source package without really understanding what is going on under the hood.
Also the way to identify the primary operator contribution of local operators is a bit naive and much better techniques have been envisage in the field (think of the Koo Saleur method and their recent implementation in the context of periodic MPS see e.g. https://arxiv.org/pdf/1710.05397.pdf
I list below some changes I would like to see before the article can be published, as mentioned I am not against publishing in Scipost Physics after the modifications, but would lean towards Scipost Core. But again I am not against Scipost Phyiscs.
Requested changes
1-Equation 2.2 define what a and b mean
2- I would add a sketch of the cylinder on which the system is described but this is just a suggestion, not mandatory.
3- Section 3 explain what folded and unfolded picture mean
4- Section 4.1 at least mention more advanced ways to map lattice operators to primary, ideally implement those more advanced ways (see above discussion)
5- Fig 3 attempt a finite size scaling analysis to understand the deviation at large distances of the Ising curve
6- In the text the position are described as x x_12 in the figures they are described as theta_1 theta_2 theta_12, please chose one notation and stay with it
7- experimental realization. The discussion about the experimental realization are pretty superficial and as it is I think could easily be omitted, since it does not add anything to the paper and seems to be added just to check one box. For example, in the case of the Rydberg architecture, the authors suggest to have a single Rydberg Hamiltonian tuned to the tricritical point in one region of the space and to the Ising critical point in the other region. The lattice system seems much simpler than the one they have described in 2.2, I would have expected some numerical simulations to validate that the two systems describe the same physics. Have they attempted such a simulation?
How important are the PBC in the scenario described and how do they imagine to impose them in the Rydberg atom setup? What would experimentalist need to measure? How can they perform such measures given their experimental capabilities?

---

## Editorial Decision

unknown